# Adenosine triphosphate is co-secreted with glucagon-like peptide-1 to modulate intestinal enterocytes and afferent neurons

Van B. Lu [1], Juraj Rievaj[1], Elisabeth A. O'Flaherty[1], Christopher A. Smith[1], Ramona Pais[1], Luke A. Pattison[2], Gwen Tolhurst[1], Andrew B. Leiter[3], David C. Bulmer[2], Fiona M. Gribble [1] & Frank Reimann [1]

Enteroendocrine cells are specialised sensory cells located in the intestinal epithelium and generate signals in response to food ingestion. Whilst traditionally considered hormone-producing cells, there is evidence that they also initiate activity in the afferent vagus nerve and thereby signal directly to the brainstem. We investigate whether enteroendocrine L-cells, well known for their production of the incretin hormone glucagon-like peptide-1 (GLP-1), also release other neuro-transmitters/modulators. We demonstrate regulated ATP release by ATP measurements in cell supernatants and by using sniffer patches that generate electrical currents upon ATP exposure. Employing purinergic receptor antagonists, we demonstrate that evoked ATP release from L-cells triggers electrical responses in neighbouring enterocytes through $P2Y_2$ and nodose ganglion neurones in co-cultures through $P2X_{2/3}$-receptors. We conclude that L-cells co-secrete ATP together with GLP-1 and PYY, and that ATP acts as an additional signal triggering vagal activation and potentially synergising with the actions of locally elevated peptide hormone concentrations.

[1] Metabolic Research Laboratories, Wellcome Trust MRC Institute of Metabolic Science, University of Cambridge, Addenbrooke's Hospital, Hills Road, Cambridge CB2 0QQ, UK. [2] Department of Pharmacology, University of Cambridge, Cambridge CB2 1PD, UK. [3] Department of Medicine, University of Massachusetts Medical School, Worcester, MA, USA. These authors jointly supervised this work: Fiona M. Gribble, Frank Reimann. Correspondence and requests for materials should be addressed to F.M.G. (email: fmg23@cam.ac.uk) or to F.R. (email: fr222@cam.ac.uk)

Enteroendocrine cells (EECs) are specialized hormone-releasing cells scattered along the gastrointestinal epithelium. In response to various stimuli following food ingestion, they release a host of gut peptide hormones, including glucagon-like peptide 1 (GLP-1), which is secreted from a sub-population of EECs traditionally called L-cells, that at least in the distal intestine often co-secrete peptide YY (PYY)[1]. GLP-1 acts as an incretin hormone, boosting glucose dependent insulin release from pancreatic β-cells and both GLP-1 and PYY suppress food intake[1]. The anorexic action of these hormones is thought at least in part to be mediated through activation of their cognate G-protein coupled receptors (GLP1R and NPY2R, respectively) located on vagal afferent nerve terminals, originating from neurons with somata in the nodose ganglia[2].

We showed previously that GLP-1 application in isolation did little to cytosolic $Ca^{2+}$-concentrations in *Glp1r*-expressing neurons isolated from nodose ganglia, consistent with the known predominant $G_s$-coupling of GLP1R; however, co-application of GLP-1 with adenosine triphosphate (ATP), a strong activator of these neurons, resulted in significantly stronger $Ca^{2+}$-signals than ATP application alone[3]. Although EECs are best recognised for their expression and release of peptide hormones, some EECs are known to produce and release small molecule transmitters such as the biogenic amine serotonin, the principal product of enterochromaffin (EC) cells, a subclass of EECs best known for their regulation of gut motility[4]. In this study, we investigate whether L-cells exhibit regulated secretion of ATP. We demonstrate that this purinergic transmitter is present in vesicles of GLP-1 secreting cells, is released in response to stimulation of enteroendocrine cells, and can signal locally to enterocytes and neurons, thus expanding our understanding of the local communication between EECs and neighbouring cells.

## Results

### ATP is located in distinct punctae in GLP-1 secreting cells.

To investigate whether ATP is accumulated in a vesicular compartment within GLP-1 secreting cells we incubated GLUTag cells, a model of murine colonic GLP-1 secreting L-cells[5], with the fluorescent dye quinacrine. Quinacrine has been shown to exhibit bright fluorescence in the presence of high concentrations of ATP[6,7] and consistent with a vesicular pool of ATP, we observed bright fluorescent punctae after 20 min incubation at 5 μM quinacrine (Fig. 1a, b). Live-cell quinacrine staining was also performed on mixed primary colonic cultures from transgenic mice in which proglucagon expressing cells are tagged with a red fluorescent protein (RFP)[8]. RFP-positive cells with the morphology of L-cells displayed punctate quinacrine staining similar to that observed in GLUTag cells (Fig. 1c, d).

Using time-lapse total internal reflection fluorescence (TIRF) microscopy to monitor changes in quinacrine fluorescence in GLUTag cells[9], we observed transient increases in the fluorescence of individual quinacrine-stained structures followed by an outwardly-spreading fluorescence (Fig. 1e–g, Supplementary Movie 1), suggestive of vesicular fusion and content release. The transient increase in fluorescence likely reflects either an increase in quantum yield due to the rapid pH/concentration change upon fusion[10] or deeper penetration of vesicles into the evanescent field[9]. These findings suggest that ATP in L-cells was concentrated in vesicles that exhibited sporadic fusion with the surface membrane.

To investigate whether ATP was contained in GLP-1 containing vesicles, we first attempted to perform co-staining for quinacrine and GLP-1, but the permeabilisation and fixation protocols needed for immunofluorescence staining were incompatible with quinacrine staining for ATP. Instead, we examined localisation of the vesicular nucleotide transporter (VNUT), the protein responsible for packaging ATP into secretory vesicles[11]. In both GLUTag cells and L-cells (Fig. 1h–j), VNUT-immunoreactivity was localized in discrete punctae of which only a minority also stained positive for GLP-1 (~8%) in GLUTag. Considerable overlap was, however, observed between VNUT and GLP-1 in primary mouse (mean ± SEM: 40 ± 6% of VNUT + ve staining positive for GLP-1, n = 19 cells) and human (54 ± 7% of VNUT + ve staining positive for GLP-1, n = 17 cells) L-cells and this was even more pronounced when co-staining for PYY was analysed (91 ± 3% (n = 19) and 91 ± 4% (n = 18) of VNUT + ve staining cells also stained positive for PYY in mouse and human, respectively).

### ATP is released from GLP-1 secreting cells.

To determine whether ATP is released from GLP-1 secreting cells, a bioluminescence assay was used to measure ATP in supernatants of cultured GLUTag cells. In control experiments assessing the stability of exogenous ATP applied onto plated GLUTag cells (Supplementary Figure 1a), levels of ATP decayed rapidly over time with 88% loss over 1 h, as also previously reported by others[12]. The loss of exogenous ATP was not due to spontaneous breakdown as there was almost full recovery in control wells that did not contain GLUTag cells (Supplementary Figure 1a, open symbols), suggesting that a cell-dependent mechanism was responsible for the decline in ATP levels. Inhibition of ectonucleotidases with polyoxotungstate (POM-1, 100 μM)[13], but not ARL-67156 (100 μM)[14], slowed the rate of ATP disappearance (Supplementary Figure 1a); in the light of these control experiments, assessments of ATP release were made after 10 min to limit ATP losses. ATP concentrations in GLUTag cell culture supernatants were significantly elevated following cell stimulation with a mixture of forskolin, IBMX and high glucose (Fig. 2a) and similar results were obtained in the absence of ectonucleotidase inhibition (Supplementary Figure 1b).

For follow-on experiments we required more selective methods to stimulate L-cell secretion, and therefore tested the activation of the angiotensin receptor AT1A, which we have previously reported to trigger L-cell selective activation in mouse and human colonic epithelium[15]. Angiotensin-II (AngII, 1 μM) significantly increased ATP concentrations in supernatants of GLUTag cells (Fig. 2a, Supplementary Figure 1b) and primary human colonic cultures (Fig. 2e). As an alternative approach to achieve cell-specific stimulation, we transiently expressed Gq-DREADD in GLUTag cells and showed that clozapine-N-oxide (CNO, 10 μM) similarly increased ATP release into the supernatants (Fig. 2b, Supplementary Figure 1c). CNO application to untransfected GLUTag cells resulted in ATP levels similar to basal conditions (Fig. 2b, Supplementary Figure 1c) confirming specificity of the DREADD agonist. We thus used this approach to selectively stimulate L-cells in primary colonic cultures derived from mice with Gq-DREADD expression under the control of the proglucagon promoter; CNO (10 μM) significantly increased supernatant ATP concentrations in the absence or presence of co-stimulating forskolin/IBMX (10 μM each; Fig. 2c). The same secretion supernatants were assayed for total GLP-1, revealing that CNO (10 μM) also increased GLP-1 release in the absence or presence of co-stimulating forskolin/IBMX (10 μM each, Fig. 2d). These findings support the idea that ATP is released from regulated secretory vesicles in L-cells.

To monitor ATP release in real time, we used sniffer patches[16], composed of slowly-inactivating $P2X_2$ receptors[17] over-expressed in HEK293 cells, as a local ATP sensor. Sniffer patches exhibited currents that were evoked by exogenous ATP, but not adenosine or ADP (Supplementary Figure 2a,b), and were blocked by

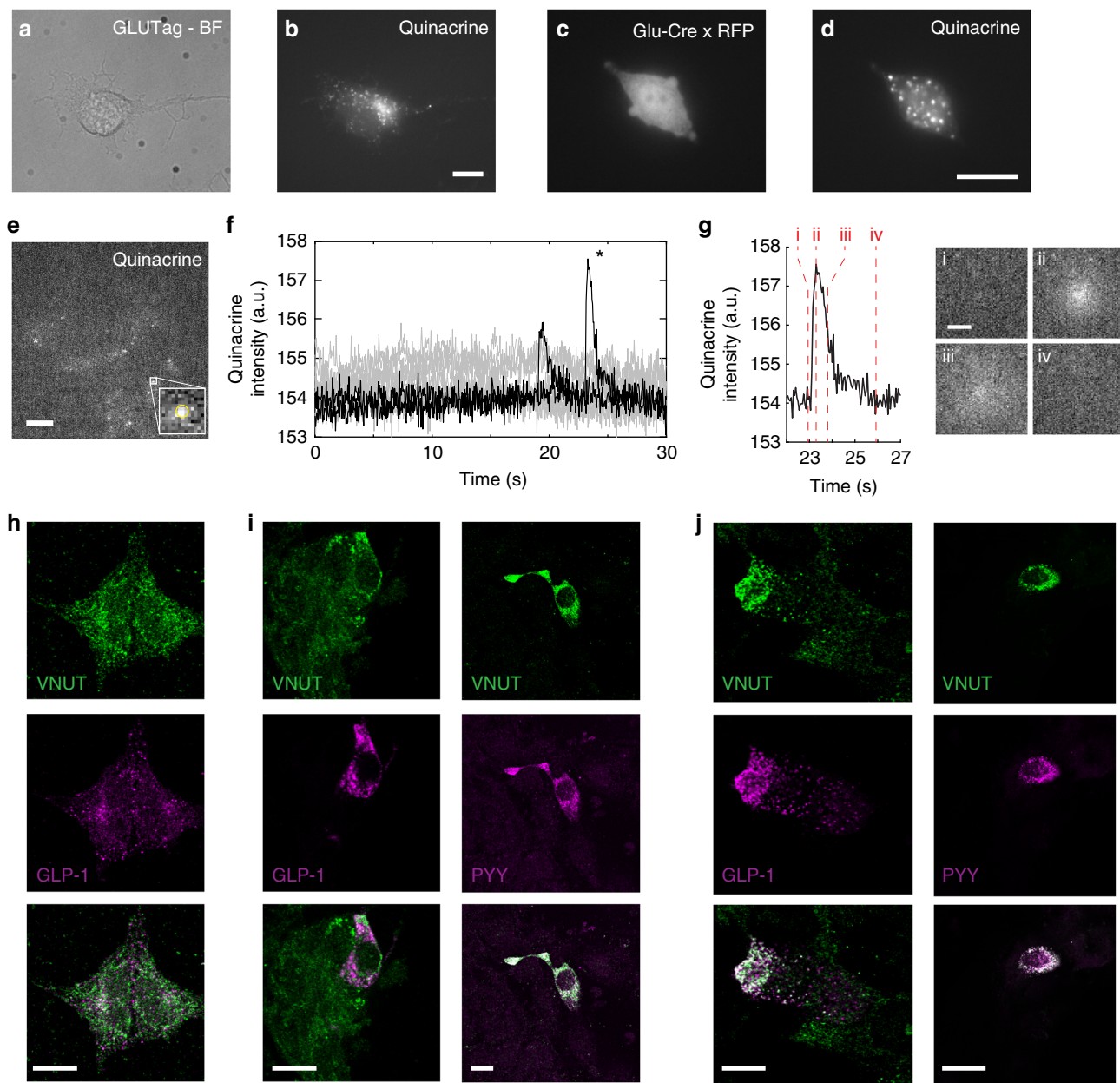

**Fig. 1** Quinacrine fluorescence and VNUT staining of GLP-1 secreting cells. **a** GLUTag cell under brightfield illumination and **b** 480 nm fluorescence excitation after incubation with 5 μM quinacrine. **c** RFP fluorescence of an L-cell from a Glu-Cre x Rosa26tdRFP reporter mouse and **d** 480 nm fluorescence of the cells shown in **c** after incubation with 5 μM quinacrine. **e** Example image from total internal reflection fluorescence (TIRF) microscopy of GLUTag cells after incubation with quinacrine (5 μM). Images were collected every 50 ms for 30 s. Inset shows zoomed view of vesicle in the white box, the yellow circle is the mask within which intensities were measured. **f** Representative profiles of quinacrine intensity for the image in **e** during transient fluorescence increase events (black) and failed secretion (grey). Transient increases occurred in 21 out of 35 cells examined with a median frequency of 14.9 spikes $mm^{-2} s^{-1}$ in responsive cells (IQR: 5.0–27.3 spikes $mm^{-2} s^{-1}$, $n = 24$ movies from seven independent experiments). **g** Intensity profile of the vesicle labelled * in **e** and **f**, with images showing vesicle signal at various time points during the spike: i = prior to spike, ii = peak of the spike, iii = during fluorescence dissipation, iv = return to initial signal. **h** GLUTag cells, **i** mouse ileum (left) or colon (right) and **j** human colonic cultures immunostained for the vesicular nucleotide transporter (VNUT, green, top panel) and GLP-1 or PYY (magenta, middle panel). Bottom panel represents merged images. Scale bars represent 10 μm, apart from **g** (3 μm)

suramin (Supplementary Figure 2c,d), a broad-spectrum inhibitor of P2X and P2Y receptors. A mixture of forskolin, IBMX and glucose or high K+ solution triggered ATP currents when sniffer patches were placed adjacent to GLUTag cells, but had no effect on sniffer patches in the absence of a local GLUTag cell (Fig. 3a, b). Sniffer patches in the vicinity of Gq-DREADD-transfected GLUTag cells exhibited currents triggered by CNO (10 μM) or AngII (1 μM) (Fig. 3c, d). Using AngII as an L-cell-selective stimulus in mouse mixed primary colonic cultures[15], ATP currents were detected from sniffer patches placed in close proximity to identified fluorescent primary colonic L-cells following AngII stimulation (Fig. 3e, f). ATP evoked currents were also detected by sniffer patches placed in close proximity to primary L-cells following stimulation by various agonists of nutrient-sensing receptors expressed on L-cells including AMG-1638 (10 μM), a selective free-fatty acid receptor 1 (FFA1)

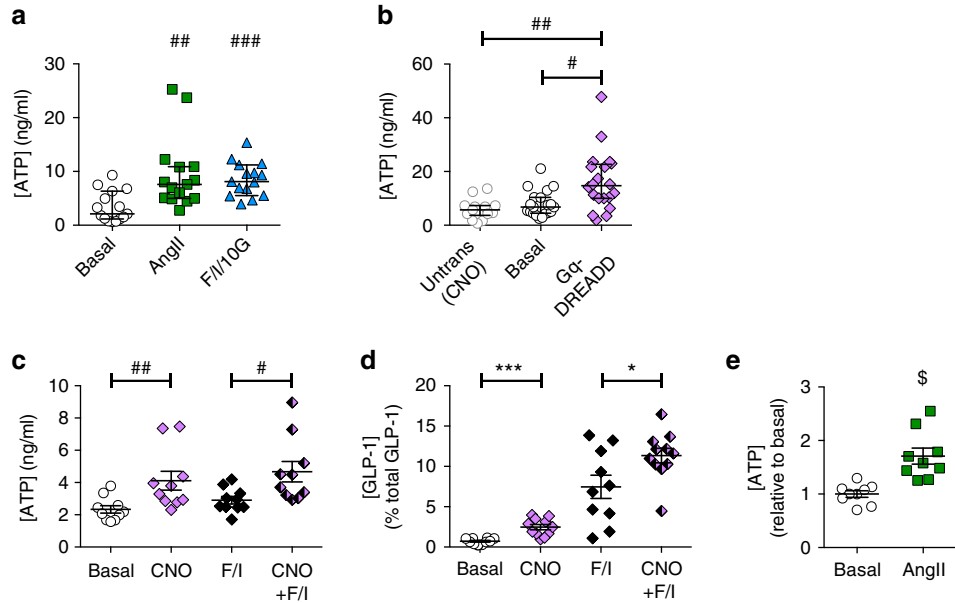

**Fig. 2** Measurements of ATP release from GLP-1 secreting cells. **a** ATP levels measured in GLUTag cell supernatants after stimulation for 10 min with AngII (1 μM) or forskolin (10 μM) + IBMX (10 μM) + 10 mM glucose (F/I/10 G); $n = 15$ wells, data obtained from five independent experiments. **b** ATP levels in supernatants of GLUTag cells transiently transfected with GqDREADD, incubated for 10 min with or without CNO (10 μM); $n = 14$–20 wells from four independent experiments. **c** ATP levels in supernatants from primary colonic cultures derived from GLU-Cre/GqDREADD mice, treated with CNO (10 μM), and forskolin/IBMX (F/I, 10 μM each) as indicated. **d** GLP-1 secretion from primary colonic cultures used in **c**, expressed as percentage total GLP-1 content in the cells. $n = 10$ wells for conditions without CNO and $n = 11$ wells for conditions with CNO from three independent experiments for **c** and **d**. **e** ATP levels measured in primary human colonic cultures after stimulation for 60 min with AngII (1 μM); $n = 9$, data obtained from four independent experiments. All experiments were performed in the presence of 100 μM POM-1 (GLUTag, **a**, **b**) or 100 μM POM-1 and 100 μM ARL-67156 (primary cultures, **c**–**e**). Individual points represent measurements from single wells and lines on the graph represent median ± interquartile range for **a**–**c** and mean ± SEM for **d** and **e**. *$p < 0.05$, ***$p < 0.001$ by one-way ANOVA with Dunnett's multiple comparisons test. #$p < 0.05$, ##$p < 0.01$, ###$p < 0.001$ by Kruskal–Wallis ANOVA with multiple comparisons test. $$p < 0.05$ by unpaired $t$-test

agonist, GPBAR-A (10 μM), a selective agonist of the bile acid receptor GPBAR1, and peptones (3 mg ml$^{-1}$; Fig. 3f).

**Paracrine modulation of colonic mucosal function by ATP.** There are several purinergic receptors expressed in mouse intestinal mucosa, with P2Y receptors in particular responsible for modulating transepithelial ion and water movement[18,19]. To assess the potential effects of ATP released from L-cells on surrounding enterocytes and the regulation of colonic mucosal functions we utilised Ussing chambers and measured the short-circuit current ($I_{sc}$) as a marker of electrogenic ion transport across the colonic epithelium. Our previous studies using mouse distal colon demonstrated biphasic responses to AngII, with $I_{sc}$ exhibiting a transient increase followed by a more sustained inhibition. Only the sustained response was sensitive to NPYR inhibition and therefore attributable to PYY[15], whereas both components were blocked by the angiotensin receptor inhibitor candesartan, suggesting that they occurred downstream of AT1A activation in L-cells. On the basis of our data demonstrating ATP secretion by L-cells, we hypothesised that the transient increase in $I_{sc}$ might reflect a response to locally-released ATP.

We were concerned that in our previous study, we only observed the transient increase in $I_{sc}$ in ~50% of preparations after AngII application, and therefore aimed to minimise this variability by removing the entire submucosal layer as well as the tunica muscularis mucosae from our colonic preparations (Fig. 4a, right). This did not diminish the ability of epithelium to generate a transepithelial potential difference, and produced consistent measurements of transepithelial resistance (TER), as well as basal $I_{sc}$ and peak $I_{sc}$ induced by forskolin at the end of each

experiment (Fig. 4b,c). Consistent with the idea that the AngII stimulated $I_{sc}$ increase might reflect a response to ATP, basolateral application of exogenous ATP (50 μM) to mucosal preparations itself induced a transient increase in $I_{sc}$ (Fig. 4d). Importantly, AngII triggered a transient increase in $I_{sc}$ in all colonic epithelial preparations in which the submucosal layer had been removed (5/5, Fig. 4e).

L-cell secreted GLP-1 was not responsible for the AngII-induced increase in $I_{sc}$ as preincubation with the GLP1R antagonist exendin-9 (1 μM) did not significantly alter the peak $I_{sc}$ amplitude (Supplementary Figure 3). However, pretreatment with suramin (100 μM) significantly impaired the AngII-induced increase in $I_{sc}$ (Fig. 4g, i). P2Y$_2$ receptors appeared to be involved since the more selective P2Y$_2$ receptor antagonist AR-C 118925XX (AR-C, 5 μM) also significantly reduced the AngII-induced peak $I_{sc}$ (Fig. 4h, i). Another broad-spectrum ATP receptor blocker PPADS (100 μM) also reduced the mean $I_{sc}$ triggered by AngII, but this did not reach statistical significance (Fig. 4f, i). After the transient $I_{sc}$ increase, AngII produced a slow, longer lasting depression in $I_{sc}$ (Fig. 4e–h), as seen previously using colonic epithelial preparations with an intact muscularis mucosae layer. This depression in $I_{sc}$, previously attributed to PYY[15], was not altered by pretreatment with suramin, AR-C, or exendin-9 (Fig. 4e–h, j, Supplementary Figure 3).

**ATP stimulates co-cultured nodose ganglion neurons.** Consistent with our previous report that cultured *Glp1r*-expressing nodose (ND) ganglion neurons exhibited Ca$^{2+}$ responses to ATP[3], we found that the majority of ND neurons were ATP-responsive, as measured using the Ca$^{2+}$-sensitive fluorescent

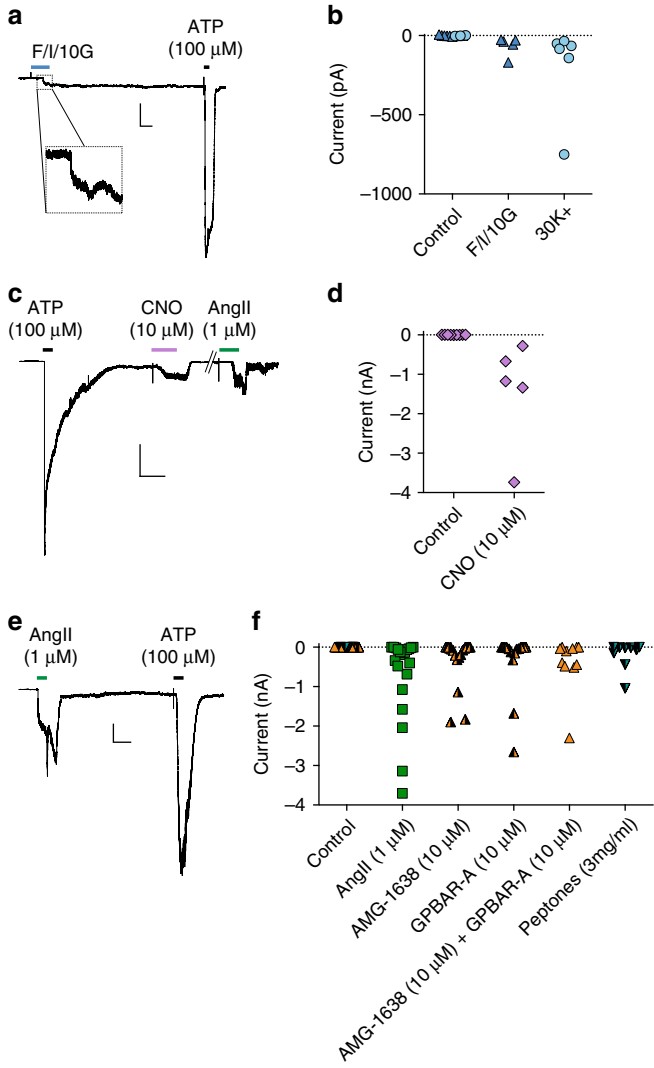

**Fig. 3** ATP secretion measured using sniffer patches. Recordings from sniffer patches excised from HEK293 cells transfected with P2X$_2$ receptors.
**a** Example trace of a sniffer patch placed adjacent to a GLUTag cell that was stimulated by forskolin/IBMX/glucose (10 μM/10 μM/10 mM; F/I/10G). 100 μM ATP was applied at the end of the recording to activate the sniffer patch directly. **b** Peak current amplitudes in response to forskolin/IBMX/glucose (blue triangles) and 30 mM K$^+$ (blue circles) from multiple cells recorded as in **a**; sniffer patches were placed either adjacent to the perfusion outlet (control—triangles and circles overlaid) or nearby a GLUTag cell. **c** Example trace of a recording from a sniffer patch placed adjacent to a GLUTag cell transiently transfected with Gq-DREADD, with addition of ATP, CNO and AngII as indicated by the bars. **d** Summary data recorded as in **c** in response to CNO from GLUTag cells either untransfected (Control) or Gq-DREADD-transfected. **e** Example trace of a sniffer patch placed adjacent to a primary colonic L-cell identified by cell-specific Venus expression driven by the proglucagon promoter. AngII and ATP were perfused as indicated. **f** Summary data from sniffer patches in response to various stimuli of endogenous receptors of L-cells, when placed either adjacent to the perfusion outlet (control—symbols overlaid) or in close proximity to a primary L-cell. Various symbols represent the stimulus applied. Y-scale bars represent 0.5 nA and X-scale bars represent 20 ms in **a**, **c**, **e**

indicator Fura-2 (202/209 ND neurons, 97%). Exogenous ATP also elevated Ca$^{2+}$ in a proportion of GLUTag cells (89/123 cells, mean ± SEM Δ340/380 ratio = 1.3 ± 0.09). To test whether ATP released from GLP-secreting cells is sufficient to activate ND

neurons, we co-cultured EYFP-labelled ND neurons (derived from NeuroD1-Cre/Rosa26-EYFP mice), together with Gq-DREADD-mCherry-transfected GLUTag cells (Fig. 5a, c) and monitored intracellular Ca$^{2+}$-dynamics in both cell types after loading with Fura-2. ND neurons cultured without Gq-DREADD-transfected GLUTag cells were not responsive to CNO (mean ± SEM Fura-2 Δ340/380 ratio = 0.08 ± 0.02, $n = 43$ neurons). In co-cultures, however, stimulation of the GLUTag cells with CNO resulted in an increase in the Fura-2 fluorescence ratio in most GLUTag cells as predicted (upper traces of Fig. 5b, d), but also elevated Ca$^{2+}$ in 30% of the co-cultured ND neurons (lower traces of Fig. 5b, d). We also observed changes in the Fura-2 ratio in response to ATP and CNO in cells present in the co-cultures that were not positive for either EYFP or mCherry. Based on their absence of a fluorescent marker, their robust ATP-responsiveness (mean ± SEM Δ340/380 ratio = 4.7 ± 0.3, $n = 26$), and their multipolar morphology, these are likely satellite-glial cells (SGCs) arising from the nodose ganglion (Supplementary Figure 4).

We used pharmacological agents to investigate whether ATP was a transmitter between GLUTag cells and ND neurons in co-cultures. Suramin (100 μM) was found not to be suitable for this experiment because it triggered a Ca$^{2+}$ response in 75% of GLUTag cells (mean ± SEM Δ340/380 ratio of responsive GLUTag cells = 0.7 ± 0.1, $n = 15$) and did not completely block exogenous ATP-triggered calcium responses in ND neurons (56 ± 5% block, $n = 22$; Fig. 6e). Instead, we used PPADS (100 μM), which did not itself trigger Ca$^{2+}$ responses in GLUTag cells or ND neurons (for GLUTag cells mean ± SEM Δ340/380 ratio = 0.07 ± 0.02, $n = 54$; for ND neurons mean ± SEM Δ340/380 ratio = −0.03 ± 0.04, $n = 27$), but blocked exogenous ATP triggered Ca$^{2+}$ elevations in ND neurons (89 ± 4% (mean ± SEM) block, $n = 27$; Fig. 6e). Gq-DREADD transfected GLUTag cells exhibited repeated Ca$^{2+}$ responses to successive applications of CNO (Fig. 5b, d), which were unaffected by PPADS (−4.7 ± 7% block, $n = 39$). Although a second application of CNO failed to induce a second increase in Ca$^{2+}$ in half of the responsive ND neurons, in ND neurons that exhibited a second response to CNO, PPADS reduced the peak area of the CNO-induced Ca$^{2+}$ rise by 53%, supporting the idea of ATP-mediated signalling from GLUTag cells to neurons (Fig. 5d–f). PPADS also partially blocked GLUTag cell responses to exogenously applied ATP (38 ± 15% block, $n = 47$) and had inconsistent effects on the satellite-glial cells that responded to CNO application (Supplementary Figure 4).

We further co-cultured ND neurones with primary cultures from transgenic mice expressing Gq-DREADD under the control of the proglucagon promoter, thus enabling cell-specific stimulation of L-cells (Fig. 5g). Ca$^{2+}$ responses were elicited in ND neurons following activation of L-cells with CNO (Fig. 5g), and were consistently reduced by PPADs application (100 μM) (Fig. 5g, h). PPADs reduced the peak amplitude of CNO-induced Ca$^{2+}$ response by 68% (Fig. 5i).

**P2X$_3$-mediates ATP signalling from L-cells to nodose neurons.** Consistent with the concept of purinergic signalling between L-cells and the afferent vagus nerve, qPCR analysis of fresh and cultured ND ganglion neurones identified expression of *P2rx2* and *P2rx3* subunits (Fig. 6a). Heterogeneity of *P2rx* subunit expression in ND neurons was evident from single-cell expression analysis (Fig. 6b); however, *P2rx3* expression was present in all ND neurons examined and its levels were the highest compared with all other *P2rx* subunits. Immunostaining for P2X$_3$ in dissociated ND cultures confirmed protein expression in GLP1R negative (Fig. 6c) and positive (Fig. 6d) neurons. To examine the

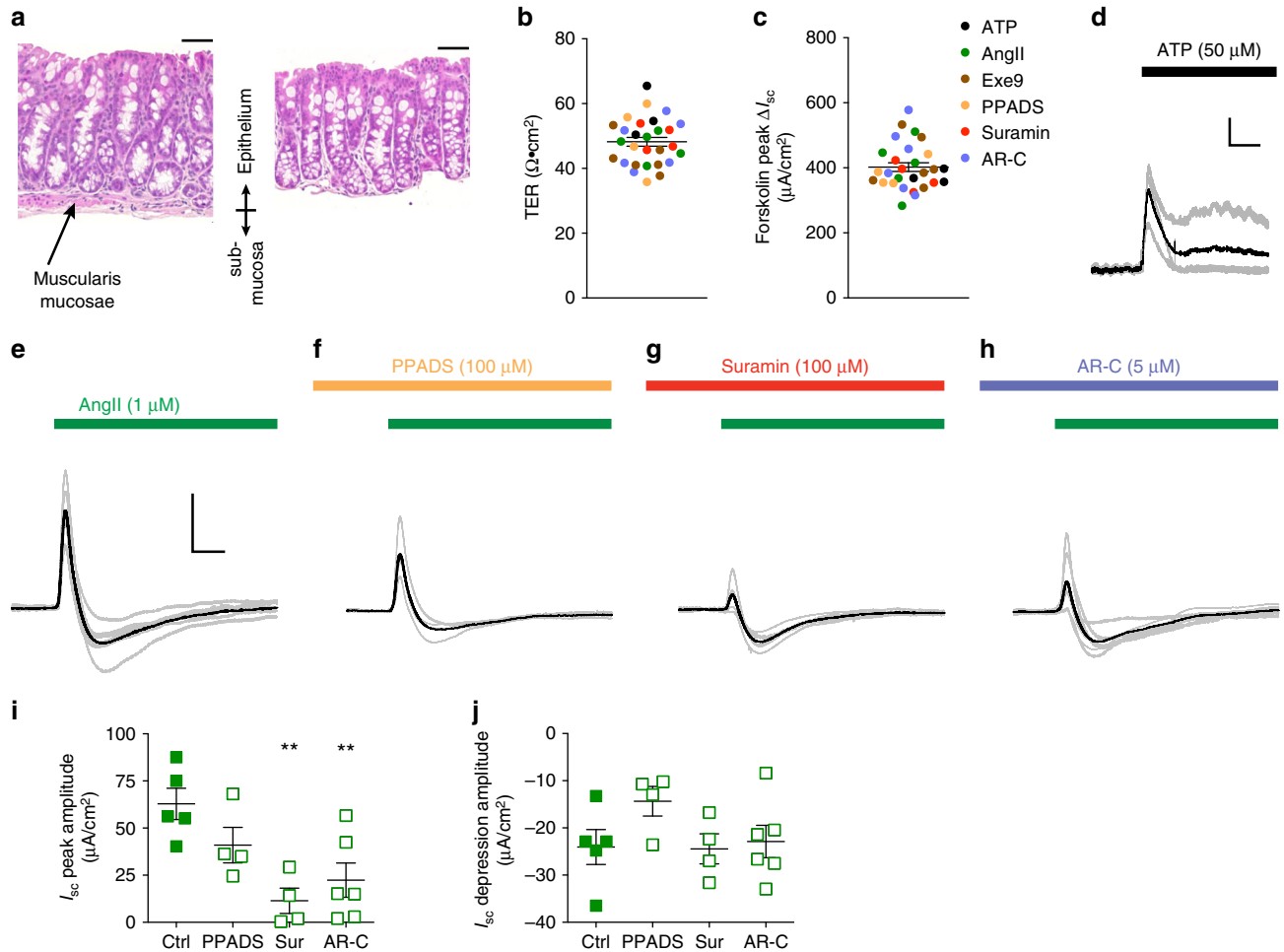

**Fig. 4** Ussing chamber recordings from murine colonic mucosa. **a** Haematoxylin and eosin stained colonic epithelium following removal of the serosa and outer muscular layer (left) or additional removal of the submucosal and muscularis mucosal layer (right); scale bars represent 50 μm. Distribution of **b** transepithelial resistance (TER) and **c** peak $I_{sc}$ amplitude induced by bilateral application of forskolin (10 μM) at the end of each experiment, colour coded according to the test drug added earlier in the experiment (coefficient of variation, CV TER = 15%, CV basal $I_{sc}$ = 28.2%, CV forskolin peak $I_{sc}$ = 17.5%). Individual data points represent individual epithelial preparations and lines represent mean ± SEM ($n$ = 25). **d** $I_{sc}$ recordings during addition of ATP (50 μM) to the basolateral compartment. Grey traces represent individual experiments and the black trace represents the mean of $n$ = 3 recordings. Mean ± SEM $\Delta I_{sc}$ = 29.5 ± 6 μA cm$^{-2}$. Y-scale and X-scale bar represent 5 μA and 200 ms, respectively. **e–h** Superimposed $I_{sc}$ recording traces from individual recordings during basolateral application of AngII (1 μM). Purinergic receptor inhibitors were added to the basolateral compartment in **f–h** for 10–12 min prior to AngII addition, as indicated above the traces. Grey traces indicate individual recordings and black traces represent the mean response. Y-scale and X-scale bar represent 10 μA and 2 min, respectively. **i, j** Amplitudes of the positive peak $I_{sc}$ (**i**), and the $I_{sc}$ depression (**j**), for the recordings depicted in **e–h**. Data points represent individual epithelial preparations and lines represent mean ± SEM. Control (Ctrl, $n$ = 5), PPADS (100 μM, $n$ = 4), Suramin (100 μM, Sur, $n$ = 4) and AR-C 118925XX (5 μM, AR-C, $n$ = 6). Statistical analysis performed using one-way ANOVA and Dunnett's multiple comparisons test, **$p$ < 0.01

functional contribution of P2X₃ in signalling between L-cells and vagal afferents, the more selective P2X₂/P2X₃ blocker Ro51 was tested on co-cultures of Gq-DREADD transfected GLUTag cells and ND neurons (Fig. 6f). GLP1R-positive ND neurons were also examined using the GLP1R-Cre mouse line[3] to identify GLP1R-expressing ND neurons. Ro51 reduced the peak amplitude of CNO-induced Ca²⁺ responses in most ND neurons (Fig. 6g) and overall inhibited CNO-triggered Ca²⁺ elevations by 54% (Fig. 6h), thus supporting the role of P2X₃ in ATP signalling between L-cells and vagal afferent neurons.

**Signalling from L-cells to sensory neurones in intact colon.** To examine whether L-cell-released ATP triggers afferent nerve signalling within the intact gut, we measured changes in mesenteric nerve activity from the proximal colon following AngII mediated L-cell activation. Reproducible biphasic increases

in nerve discharges were elicited by bath application of AngII (1 μM) following pretreatment with IBMX (100 μM; Supplementary Figure 5a, b, f). This consisted of a rapid transient increase in nerve firing followed by a sustained plateau of activity lasting more than 10 min. Repetitive AngII responses could be obtained from the same sample with similar response profiles and minimal desensitization (Supplementary Figure 5c, d, e). No significant change was observed in the transient response in the presence of a purinergic antagonist, whilst the plateau phase of AngII responses was largely attenuated following pre-treatment with PPADS (Supplementary Figure 5e, g, h).

## Discussion
Beyond its roles as an energy source for numerous biochemical processes and a stabilizer of catecholamine loading in secretory vesicles[20], ATP has been widely regarded as a signalling molecule

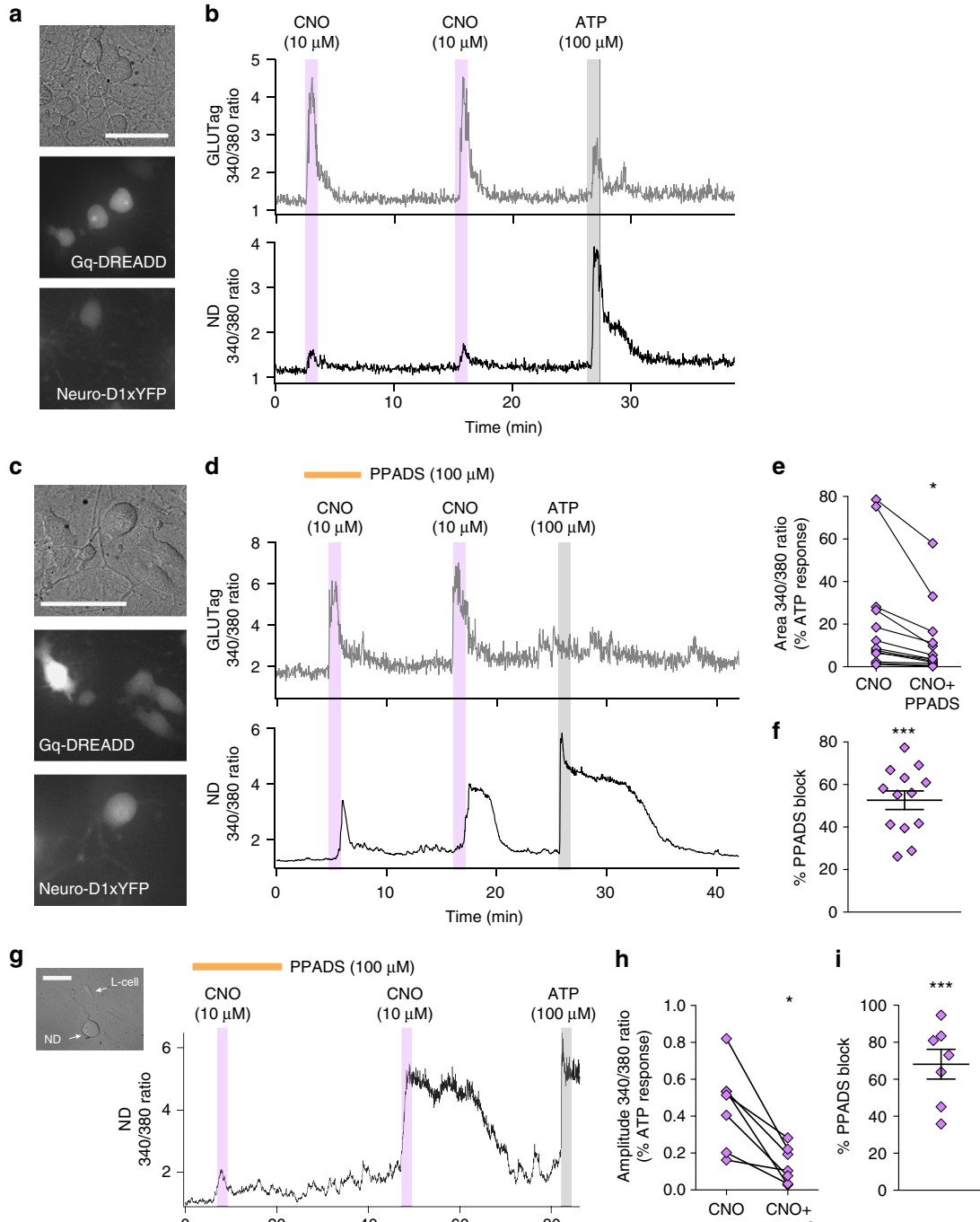

**Fig. 5** $Ca^{2+}$ imaging of nodose (ND) neuron and Gq-DREADD-transfected GLUTag co-cultures. **a**, **c** Images of cells in our co-culture system: brightfield (top), fluorescence at 550 nm excitation (middle), and fluorescence at 488 nm excitation (bottom). Scale bars represent 50 μm. **b**, **d** Intracellular $Ca^{2+}$ levels, represented as the ratio of Fura-2 fluorescence at 340 and 380 nm excitation, of the cells shown in **a** and **c**: mCherry-positive Gq-DREADD transfected GLUTag cell (top trace) and YFP-positive ND neuron (lower trace) identified using a NeuroD1-Cre/YFP reporter mouse. Drugs were applied as indicated above the traces. **e** Areas under the curve of the Fura-2 ratio change of ND neurons in response to CNO (10 μM) with and without PPADS (100 μM) pre-treatment. Values are expressed relative to the area of the response induced by ATP (100 μM) in each cell. Only ND neurons where a $Ca^{2+}$ response could be evoked by CNO after wash out of the PPADS were included. Individual data points represent each ND neuron ($n = 13$). Statistical analysis performed using a paired $t$-test, $*p < 0.05$. **f** Percentage block of CNO responses in ND neurons by 100 μM PPADS, derived from the data shown in **e**. Individual data points represent each ND neuron and lines represent mean ± SEM ($n = 13$). Statistical analysis performed using one-sample $t$-test, $***p < 0.001$. **g** Fura-2 imaging of ND neuron cultured with primary L-cells expressing Gq-DREADD under the control of the proglucagon promoter. Inset image to left is a brightfield image of the co-culture system. Drugs were applied as indicated above the traces. **h** Amplitude of Fura-2 ratio changes of ND neurons in response to CNO (10 μM) with and without PPADS (100 μM) pre-treatment. Values are expressed relative to the max response elicited by ATP (100 μM) in each cell. Individual data points represent each ND neuron ($n = 7$). Statistical analysis performed using a paired $t$-test, $**p < 0.01$. **i** Percentage block of CNO responses in ND neurons by 100 μM PPADs, derived from the data shown in **h**. Individual data points represent each ND neuron and lines represent mean ± SEM ($n = 7$). Statistical analysis performed using one-sample $t$-test, $***p < 0.001$

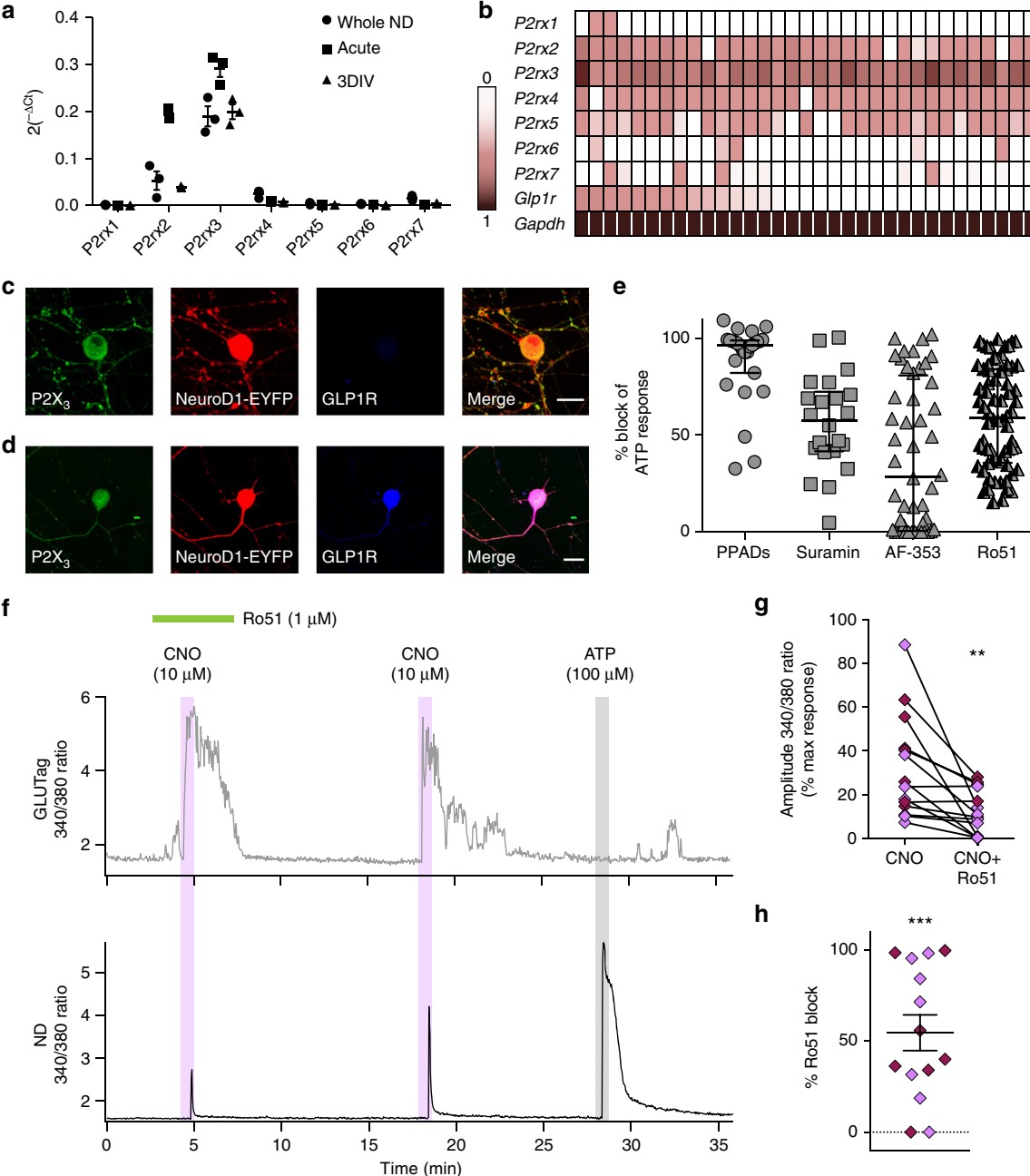

**Fig. 6** Expression analysis and pharmacological characterization of ATP receptors in nodose (ND) ganglion neurons. **a** *P2rx* subunit expression levels ($2^{-\Delta Ct}$ values) of ND neurons from intact ganglia (black circles), acutely dissociated neurons (black squares), and after 3 days in vitro cultures (black triangles). Samples for each type of preparation were prepared from ND ganglia pooled from 2 to 3 mice, repeated three independent times. Individual data points represent independent preparations and lines represent mean ± SEM ($n = 3$). **b** Heat map of *P2rx* subunit expression from individually picked ND neurons. Each column represents a single ND neuron. Range indicator for heat map on left. Sample GLP1R negative (**c**) and GLP1R-positive (**d**) NeuroD1-EYFP neuron immunostained for P2X$_3$ (Alomone P2X$_3$ antibody APR-016 in **c**, Neuromics P2X$_3$ antibody GP10108 in **d**) and GLP1R. Scale bars represent 20 μm. **e** Scatterplot of % block of exogenous ATP (100 μM) application by 100 μM PPADs (grey filled circles, $n = 27$), 100 μM suramin (grey filled squares, $n = 22$), 30 μM AF-353 (grey filled triangles, $n = 48$) or 1 μM Ro51 (half grey filled triangles, $n = 72$) as measured by Fura2 Ca$^{2+}$ imaging. Individual data points represent individual neurons and lines represent median ± interquartile range. **f** Intracellular Ca$^{2+}$ levels, as the measured Fura2 signal (340/380 nm ratio), of an mCherry-positive Gq-DREADD transfected GLUTag cell (top trace) and GLP1R-positive ND neuron (lower trace) identified using a GLP1R-Cre/GCaMP3 reporter mouse. Drugs were applied as indicated above the traces. **g** Amplitude of Fura-2 ratio changes of ND neurons in response to CNO (10 μM) with and without Ro51 (1 μM) pre-treatment. Values are expressed relative to the max response elicited by ATP (100 μM) in each cell. Individual data points represent each ND neuron ($n = 14$). Dark purple symbols represent identified GLP1R-positive ND neurons. Statistical analysis performed using a paired *t*-test, ** $p < 0.01$. **h** Percentage block of CNO responses in ND neurons by 1 μM Ro51, derived from the data shown in **g**. Individual data points represent each ND neuron and lines represent mean ± SEM ($n = 14$). Statistical analysis performed using one-sample *t*-test, *** $p < 0.001$

in its own right[21]. In this study, we provide evidence for regulated ATP release from enteroendocrine L-cells and demonstrate functional puringergic signalling from L-cells to enterocyte and neuronal targets within the colonic mucosa.

Following stimulation of GLUTag and primary intestinal cultures by known L-cell secretagogues, ATP secretion was detected as an increased ATP content in cell supernatants, and by local sniffer-patch recordings of evoked ATP currents. This is consistent with ATP being contained in secretory vesicles, either costored with GLP-1 or in a distinct vesicular pool. VNUT (Slc17a9), the transporter reported to underlie ATP accumulation in secretory vesicles, was detected by immunohistochemistry in GLUTag and primary L-cells, with an overlap with GLP-1 and PYY positive secretory vesicles, which was high in primary tissue derived cells. This contrasts with a previous study which was unable to detect vesicular co-localisation of VNUT with GLP-1 and concluded that the transporter might be restricted to small vesicles[12]. Consistent with the partial colocalization of ATP and GLP-1 reported here, ATP has also been detected in dense core vesicles together with insulin in pancreatic β-cells[22] and ATP is localised in dense core vesicles in chromaffin cells, where it is co-released with catecholamines[20]. Our observation that a proportion of GLUTag cells exhibited cytosolic $Ca^{2+}$ responses to exogenous ATP application might suggest a positive autocrine feedback of ATP on L-cell secretion. However, GLP-1 levels were previously reported not to be altered in VNUT-knockout mice after an oral glucose challenge[22], suggesting that L-cell vesicular ATP does not play a major role in autocrine regulation of GLP-1 release under these conditions. By contrast, in pancreatic β-cells ATP appears to act as a negative feedback regulator of insulin secretion[22], and in chromaffin cells reduced ATP content in secretory vesicles impaired the ability to concentrate catecholamines[20].

It is well known that ATP and UTP stimulate colonic chloride secretion. In Ussing chamber experiments employing rabbit, rat, mouse and guinea pig colon as well as Caco2 and T84 cell lines, basolaterally-applied ATP and/or UTP caused increases in $I_{sc}$[23–25] that were largely attributed to chloride secretion[19,26,27] triggered by a transient increase in cytoplasmic calcium concentration inside epithelial cells[28–31]. Experiments with knockout mice demonstrated roles for both P2Y$_2$ (in jejunum) and P2Y$_4$ (in jejunum and distal colon) in mediating $I_{sc}$ responses to UTP[19]. We similarly observed a transient increase in $I_{sc}$ in our Ussing chamber preparations after adding exogenous ATP or following L-cell stimulation by AngII. Responses to AngII were largely inhibited by the nonspecific P2X/P2Y blocker suramin or the more specific P2Y$_2$ antagonist AR-C118925XX, but only weakly blocked by PPADS, suggesting the involvement of P2Y$_2$[32,33]. A recent report implicated GLP-1 in a similar, albeit smaller, $I_{sc}$ elevation in response to L-cell stimuli[34]. In our hands, however, the GLP1R antagonist exendin-9 had negligible effects on either basal $I_{sc}$ or $I_{sc}$ responses to AngII (Supplementary Figure 3), indicating that GLP1R was not required for the observed responses to AngII. As previously reported responses to the GLP1R agonist exendin-4 were sensitive to tetrodotoxin, suggesting neuronal involvement[34], the lack of a GLP1R-dependent $I_{sc}$ response in our preparations might reflect the removal of the submucosal neuronal plexus. Whilst L-cell signalling to neighbouring enterocytes via PYY is believed to play a role in volume homoeostasis, the potential physiological importance of the transient $I_{sc}$ elevation attributable to L-cell released ATP is currently unclear.

Using co-cultures of GLUTag cells or primary L-cells with ND neurons, we demonstrated that neurons were activated upon cell-restricted stimulation of L-cells. This likely reflects a functional connectivity between these different cell types, as formation of

synapse-like interactions between sensory neurons and enteroendocrine L-cell basolateral extensions have been described previously[35]. A simple build-up of ATP in the medium is an unlikely explanation for our observations as the recordings were performed with relatively fast continuous perfusion (~1 ml min$^{-1}$, equalling ~3 volume replacements per minute). However, local paracrine signalling cannot be excluded, as in some cases we noticed neuronal responses that appeared delayed relative to those seen in GLUTag cells monitored in the same optical field (Fig. 5b, d). We also found that in only about half of the responsive GLUTag-ND connections, could a repeated neuronal response be elicited by a second CNO stimulus. Global ATP receptor downregulation seems an unlikely explanation for this finding because in control experiments multiple applications of ATP induced repeated neuronal $Ca^{2+}$ responses of similar amplitudes. Synaptic-like connection-selective receptor downregulation or depletion of GLUTag cell ATP containing vesicles at the site of contact with ND neurons remain as possible explanations. Nevertheless, when reproducible functional connections could be recorded between GLUTag or primary L-cells and ND neurons, blockade of ATP receptors by PPADS was sufficient to reduce the amplitude and area of the neuronal $Ca^{2+}$ responses, despite equivalent CNO-induced responses in the monitored L-cells (Fig. 5). This result suggests that ATP from L-cells acts as a neurotransmitter, capable of activating local nerve endings expressing purinergic receptors such as the sensory vagal afferent neurons of the nodose ganglion, which we found by qPCR and immunostaining to express P2rx3. As some neuronal responsiveness was preserved after purinergic blockade, however, purinergic receptors insensitive to PPADS, or other molecules secreted from GLUTag cells may participate in the communication with ND ganglion neurons in co-culture. In addition to GLP-1 and ATP, for example, GLUTag cells also secrete several other peptide hormones[36] including CCK—a known stimulus of $Ca^{2+}$-responses in ND ganglion neurones[37]—as well as glutamate[38], which has been reported to target NMDA receptors on the afferent vagus[39] and to underlie L-cell to nodose neuron signalling[40]. Whilst these in vitro results could potentially represent an artefact of the co-cultures employed, we showed that functional ATP-dependent cross-talk with afferent nerves is also observed in an acute ex vivo colonic preparation. Interestingly, it was the delayed sustained response to AngII application that was inhibited by PPADS, suggesting that ATP plays a modulatory role rather than acting as a fast neurotransmitter in this preparation.

We have recently demonstrated the presence of angiotensin receptors on colonic afferent neurons[41] and consistent with the expression of these receptors on L cells[15] and sensory nerves, including nodose ganglion neurons[42,43], a biphasic afferent response was observed in response to administration of AngII in afferent nerve recordings from intact colonic preparations. The initial phase of this response was not inhibited by PPADS in keeping with a direct activation of afferent endings by AngII, however, the late phase response was abolished by pre-treatment with PPADS in line with the indirect activation of colonic afferents by ATP. Colonic L-cells, which in contrast to other intestinal epithelial cells express high levels of Agtr1a[15], are the likely source for this ATP, although we cannot exclude contributions from other cells present in the preparation. Other L-cell derived products, such as CCK, GLP-1 or glutamate might additionally contribute to the PPADS insensitive component of the afferent nerve response.

Enteroendocrine and especially L-cells are in the spotlight as targets for the treatment of obesity and diabetes. As GLP-1 has a very short plasma half-life of ~1–3 min, it is widely believed that at least some of its anorexic action is mediated through activation of GLP1R located on afferent nerve endings in the vicinity of L-

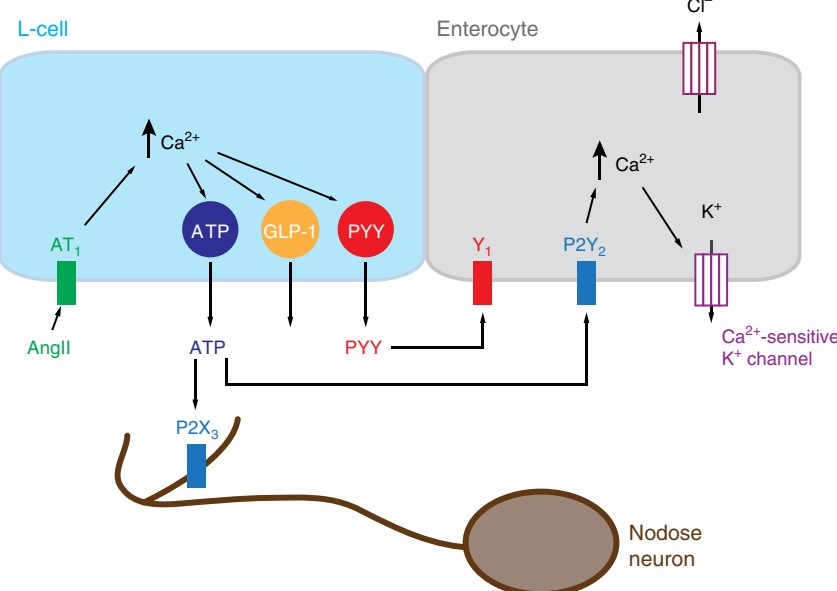

**Fig. 7** Schematic model of purinergic signalling from L-cells to local targets. ATP released together with GLP-1 and PYY from L-cells, exemplified here by angiotensin receptor stimulation, can activate purinergic receptors located on neighbouring enterocytes, potentially modulating fluid secretion by opening of $Ca^{2+}$-sensitive $K^+$ channels. ATP released from L-cells also targets local nerve terminals of vagal afferent neurons deriving from the nodose ganglion (brown cell) which express purinergic $P2X_3$ receptors, providing ATP neurotransmitter mediated signalling pathway from the intestinal epithelium to the brainstem

cells. Consistent with this idea, subdiaphragmatic vagal deafferentation[44] and selective knock-down of *Glp1r* in the nodose ganglion[45] have been shown to impair peripheral GLP1R-agonist induced reduction of food intake in rats. Recent research surprisingly demonstrated that serotonin-secreting enterochromaffin cells were the major chemical sensors for vagal afferent neurons in the proximal duodenum, whereas *Glp1r*-expressing ND neurons predominantly innervated muscle layers of the stomach and responded to stretch rather than to chemical stimuli entering the duodenum[46]. Further research suggested that GLP-1 might nonetheless activate small intestinal enterochromaffin cells, which have been reported to lack common nutrient-sensing G-protein coupled receptors, but do express *Glp1r*[47]. Our observation that L-cells not only release GLP-1 and PYY but also secrete ATP, which, as re-established here, is a potent activator of ND neurons, has implications for how we perceive the interplay between enteroendocrine cells, as primary sensors located in the epithelium layer and their secondary neuronal targets. The secretion of neurotransmitters like ATP from L-cells has the capability not only of triggering rapid signalling to local neurons, glia and enterocytes (Fig. 7), but also of synergising with the locally elevated concentrations of peptide hormones including GLP-1, that are co-released with ATP from enteroendocrine cells.

## Methods

**Tissue preparations**. All animal procedures were approved by the University of Cambridge Animal Welfare and Ethical Review Body and were conducted in accordance with the UK Animals (Scientific Procedures) Act 1986 Amendment Regulations (SI 2012/3039). The animal work was performed under the UK Home Office Project License 70/7824 and male and female mice aged 3–6 months, on a C57BL/6 background were used in this study. All human research in this study was conducted in accordance with the principles of the Declaration of Helsinki and after approval by Cambridge University Hospitals R&D and the Cambridge Central Research Ethics Committee (Ref: 09/H0308/24). All participants gave informed written consent.

To identify GLP-1 producing EECs, transgenic mice expressing Venus[48] or Cre[8] under the control of the proglucagon promoter and/or suitable Cre-reporters (tdRFP[49] or Gq-DREADD (Jax-stock # 026220)) were used. Colonic crypts were isolated and cultured from 3 to 6-months-old mice. Mice were killed and the colon was excised. Luminal contents were flushed with cold PBS and the outer muscle

layer was removed. Tissues were minced and digested with collagenase type XI (0.35 mg ml$^{-1}$) and plated onto Matrigel (BD Bioscience) coated 35 mm plastic dishes, glass-bottom dishes or glass coverslips. Mixed primary cultures of human colon were prepared from anonymised patient biopsy samples using a similar method as described above.

To identify nodose (ND) neurons in co-culture experiments, nodose ganglia were collected from a transgenic mouse line, generated by crossing NeuroD1-cre mice (provided by Andrew Leiter) with a ROSA26EYFP reporter line. Some experiments were performed with neurons isolated from GLP1R-Cre crossed with ROSA26GCaMP3 mice[3]. Nodose ganglia were isolated and placed in chilled Hanks balanced salt solution. Ganglia were digested with collagenase A (1.5 mg ml$^{-1}$) and trypsin (0.20 mg ml$^{-1}$) for 1 h at 37 °C and mechanically dissociated with fire-polished Pasteur pipettes. Neurons were centrifuged ($60 \times g$ for 6 min) and resuspended in Dulbecco's modified Eagle medium (DMEM) supplemented with 10% FBS, 100 units ml$^{-1}$ penicillin, 0.1 mg ml$^{-1}$ streptomycin and 50 ng ml$^{-1}$ nerve growth factor (NGF). Experiments were performed 48–72 h after dissociation (48 h for co-cultured GLUTag cells or 72 h for co-cultured primary L-cells).

**Cell line culture**. GLUTag cells (a kind gift from Dan Drucker, Toronto) were maintained in Matrigel-coated flasks and low-glucose DMEM supplemented with 10% FBS, 100 units ml$^{-1}$ penicillin and 0.1 mg ml$^{-1}$ streptomycin. HEK293A cells (Q-BIOgene) were maintained according to manufacturer's instructions (ATCC). Cells were transfected using Lipofectamine 2000 (Thermo Fisher) following the manufacturer's instructions. Cells for experiments were plated on 1% Matrigel-coated 24-well plates for secretions, glass-bottom dishes for imaging, or glass coverslips for sniffer patch recordings.

**Staining**. Hematoxylin and eosin staining: Sections were fixed with 4% paraformaldehyde and embedded in paraffin, sectioned into 2–5 µm-thick slices and mounted on SuperFrost slides before sections were stained with hematoxylin and eosin. Slides were imaged using a ×20/0.8 NA objective and Axioscan Z1 slide scanner (Zeiss).

Live-cell staining: 24 h after plating, dissociated GLUTag cells or primary colonic mixed epithelial cultures were removed from the incubator and gently washed with standard saline (138 buffer) then incubated with 5 µM quinacrine for 20 min at room temperature. Dishes were washed with fresh saline before imaging with an Olympus IX71 microscope with a ×60/1.4 NA oil-immersion objective and an OrcaER camera (Hamamatsu), and using MetaFluor software (Molecular Devices).

Vesicle immunostaining: Plated cells were fixed with 2% paraformaldehyde for 30 min then permeabilized with 0.1% TritonX-100 for 20 min. Guinea pig polyclonal antibodies against the vesicular nucleotide transporter (VNUT; Millipore, ABN83, 1:500) and rabbit polyclonal antibodies against GLP-1 (Abcam, ab2625, 1:200) or PYY (Abcam, ab22663, 1:200) were incubated 48 h at 4 °C. Secondary antibodies conjugated to Alexa-Fluor 488 and 555 (Invitrogen,

Supplementary Table 1) were incubated at 1:300, for 1 h before mounting on microscope slides using Prolong Gold. Immunostained cells were imaged on an SP8 confocal microscope and acquisition software (LAS X, Leica Microsystems). Overlapping staining of VNUT and GLP-1 was determined using MATLAB (R2018a; MathWorks) and previously developed methods[50]. The staining in GLUTag cells was not of high enough quality for this approach—instead a mask was set to 2× background to estimate the percentage of VNUT-positive regions also staining positive for GLP-1. Nodose immunostaining: Similar methods as above were utilised except the following primary antibodies were used: P2X₃ (Alomone, APR-016, 1:200 or Neuromics, GP10108, 1:200), GFP (Abcam, ab13970, 1:2000) and GLP1R antibody[51] (0.1 mg ml$^{-1}$). Appropriate secondary antibodies conjugated to Alexa-Fluor 488, 555 or 633 (Supplementary Table 1) were incubated at 1:300 for 2 h before mounting on microscope slides using Prolong Gold. Confocal images were acquired on an SP8 confocal microscope.

**ATP secretion assay.** GLUTag cells at a density of $0.5 \times 10^6$ cells ml$^{-1}$ or primary colonic cultures were plated on Matrigel-coated 24-well plates. Cells were washed three times with standard saline (138 buffer) containing 0.1% BSA in the presence or absence of ectonucleotidase inhibitors as indicated, for 30 min at 37 °C. After pre-treatment, cells were stimulated by addition of forskolin + IBMX (10 μM each) + 10 mM glucose, or AngII (1 μM) or CNO (10 μM) for 10 min at 37 °C. After incubation, supernatants were taken, remaining cellular debris removed by centrifugation ($200 \times g$, 5 min, 4 °C), and the amount of ATP was measured using the CellTiter-Glo® 2.0 (Promega) ATP kit and a Tecan Spark plate reader.

**GLP-1 secretion assay.** Mouse primary colonic cultures used in ATP secretion experiments were also assayed for GLP-1. Lysate samples were obtained by adding 250 μl lysis buffer to each well, scraping and collecting cellular contents, followed by centrifugation of collected lysates at $8000 \times g$ for 10 min at 4 °C. GLP-1 levels were measured using the total GLP-1 ELISA kit (MesoScale) as per manufacturer instructions. GLP-1 levels in supernatants were expressed as a percentage of total GLP-1, calculated from the GLP-1 concentration measured in supernatants/(GLP-1 concentration in supernatants + lysates) × 100.

**Sniffer-patch recordings.** For sniffer patch recordings, 1–3 MΩ resistance fire-polished borosilicate glass electrodes with tips coated with refined beeswax were used and filled with internal pipette solution containing (in mM): CsCl (125), Tris-creatine-PO₄ (5.4), EGTA (10.0), HEPES (5.0), CaCl₂ (2.0), MgCl₂ (1.0), pH 7.4, 281 mOsm kg$^{-1}$. Gap-free recordings of currents were recorded in whole-cell voltage clamp mode using an Axopatch 200B connected through a Digidata 1440A A/D converter and pCLAMP software (Axon Instruments). Sniffer patches, comprised of HEK293A cells transfected with a P2X₂ expression plasmid, were plated on glass coverslips and placed within a dish containing either GLUTag cells or mixed primary colonic cells. Following establishment of whole-cell access from HEK293 cells, outside-out patches were established by carefully withdrawing electrodes away from cells. Sniffer patches were then placed near a locally placed perfusion outlet for control measurements, or nearby cells of interest (GLUTag or L-cells).

**TIRF imaging.** GLUTag cells were plated on 2% Matrigel-coated #0 glass-bottom dishes. Prior to imaging, cells were gently washed with imaging medium (138 buffer + 0.1 mM glucose) then incubated with 5 μM quinacrine for 20 min at room temperature. Dishes were washed with imaging medium, and filled to 1 ml imaging medium for imaging. TIRF movies were obtained using a Nikon Eclipse TE2000-S microscope (Nikon) and TIRF assembly (Cairn), employing a ×100/1.49 NA oil-immersion objective, QuantEM 512SC EMCCD camera (Photometrics), 491 nm laser excitation, 535/25 nm emission filter and environmental stage-top chamber (Okolab) with base and lid heated to 37 °C. Movies were acquired using stream acquisition in MetaMorph with 50 ms exposure (20 Hz) for 30 s, and saved in .tif format. Images formed a field of view of 512 × 512 pixels, with pixel size 160 × 160 nm. Analysis of movies was performed using MATLAB® (MathWorks). Movies were read into MATLAB using BioFormats (Open Microscope Environment), then vesicles exhibiting spike profiles and vesicles failing release with various levels of quinacrine intensity were manually selected, up to a total of 20–30 vesicles per movie. Vesicles were observed to remain stationary for the duration of each movie (for >99% of vesicles). Quinacrine intensity was measured at each time point within a 300-nm-radius circular mask centred at the manually selected pixel (see Fig. 1e).

**Ussing chamber recordings.** Mid-colonic sections of male mice (1.5–4.5 cm from the ileocaecal connection and distally from mucosal folds of the proximal colon) were cut open longitudinally along the remnants of the mesenteric attachment and rinsed in Ringer's solution. The serosa, longitudinal and circular muscular layer, submucosa as well as tunica muscularis mucosae were removed by fine forceps. For technical reasons, this was done in mid-colonic sections rather than the distal colon, as used previously[15]. The tissue was mounted in an Ussing chamber (EM-LVSYS-4 system with P2400 chambers and P2404 sliders, all from Physiologic Instruments, San Diego, CA, USA; active epithelial surface = 0.25 cm²), with both parts of the chamber filled with 3 ml of Ringer's solution, maintained at 37 °C and

continuously bubbled with 5% vol/vol CO₂/ 95% vol/vol O₂. The transepithelial potential difference was clamped to 0 mV using a DVC 1000 amplifier (WPI, Sarasota, FL, USA) and the resulting short-circuit current was recorded through Ag-AgCl electrodes and 3 mol l$^{-1}$ KCl agarose bridges. The recordings were collected and stored using Digidata 1440 A acquisition system and AxoScope 10.4 software (both from Molecular Devices, Sunnyvale, CA, USA). One to two preparations from each animal were used. The transepithelial resistance and short-circuit current ($I_{sc}$) were allowed to stabilize for at least 30 min before the application of drugs. During this period, transepithelial resistance was assessed by measuring current changes in response to 2 mV pulses lasting 2.5 s, applied every 100 s. After stabilizing basal $I_{sc}$ and TER, the following drugs were applied to the basolateral compartment: 50 μM ATP, 100 μM suramin, 100 μM PPADS, 1 μM Exendin 9-39, 5 μM AR-C 118925XX or 1 μM AngII. Forskolin (10 μM) was applied bilaterally at the end of each experiment to confirm viability of the tissue.

**Calcium imaging.** Co-cultures of ND neurons and GLUTag cells were loaded with 5 μM Fura-2-acetoxymethyl ester (Molecular Probes) for 30 min in standard saline and Fura-2 calcium imaging was performed with an inverted fluorescence microscope (Olympus IX71, UK) with a ×40 oil-immersion objective, coupled to a 75 W xenon arc lamp and a monochromator (Cairn Research, Faversham, UK) controlled by MetaFluor software (Molecular Devices, UK). Emission was recorded with an Orca-ER CCD camera (Hamamatsu, UK) whilst cells were continuously perfused at ~1 ml min$^{-1}$. Fura-2 was excited at 340, 360 and 380 nm, YFP at 488, mCherry at 555 nm. Fura-2 fluorescence measurements were acquired at 0.5 Hz and analysed, after background subtraction, using MetaFluor software. As a measure of $[Ca^{2+}]_i$, the fluorescence emission ratio at 340/380 excitation was calculated. The ratio was calculated on a pixel-by-pixel basis and a user-selected area marked inside the cell. The data are shown as a change in fluorescence ratio (Δ340/380 ratio) normalised to exogenous ATP responses. The baseline was taken as the average signal 20 s before drug application and the threshold for peak Ca²⁺ responses was set as a Δ340/380 ratio > 0.2. The percentage block by PPADS or Ro51 of CNO- or ATP-induced Ca²⁺ rises was calculated as the difference in signal with and without antagonists ÷ signal without antagonist ×100.

**Ex vivo mesenteric nerve recording from mouse colon.** Mice (C57BL/6, >10 weeks old) of either sex were humanely killed by cervical dislocation, and the proximal colon with associated mesentery and extrinsic innervation removed. Tissues were superfused (7 ml min$^{-1}$; 32–34 °C) with carbogenated Krebs buffer (in mM: 124 NaCl, 4.8 KCl, 1.3 NaH₂PO₄, 1.2 CaCl₂.2H₂O, 1.2 MgSO₄.7H₂O, 10 glucose, 25 NaHCO₃; pH = 7.4 ± 0.2) supplemented with atropine (10 μM) to reduce smooth muscle contraction and indomethacin (3 μM) to block endogenous prostanoid production. Suction electrode recordings[52] were made from mesenteric nerve bundles in colons cannulated as a tubular preparation and luminally perfused (100 μl min$^{-1}$, under 2–4 mmHg pressure) with the same supplemented Krebs buffer. Drugs were bath applied to the serosal surface. Cholecystokinin (CCK, 300 nM) was routinely applied to demonstrate the presence of vagal afferent fibres in the nerve bundles recorded, as previous studies by Richards et al. have shown that mesenteric nerve responses to CCK are abolished by vagotomy[53] and produced a robust increase in nerve firing activity (mean ± SEM: + 16.4 ± 3 spikes s$^{-1}$, $N = 8$). In control experiments AngII (1 μM, 30 ml) and IBMX (100 μM) was applied following pretreatment with IBMX (100 μM, DMSO vehicle). In test protocols PPADS was given at least 20 min prior to treatment (30 μM, 150 ml) and repeat AngII treatments was applied at least 1 h apart. Ongoing nerve activity was determined by averaging the spike discharge exceeding a threshold level set at twice baseline. Changes in nerve discharge following treatment were determined as subtracted from baseline (5 min prior to treatment) and normalized to the initial peak response to the first AngII treatment.

**Tissue and single-cell qRT-PCR.** RNA from nodose ganglia was extracted using the RNeasy Micro Plus Kit (Qiagen). For picking individual cells, nodose ganglia were dissociated and plated 8–24 h before manually selecting cells with a glass suction electrode that was positioned with a micromanipulator (Luigs Neumann). Nodose neurons were drawn into the glass electrode with suction and the glass electrode tip broken into a PCR tube containing CellsDirect One-Step qRT-PCR reaction mix (Invitrogen) and SUPERase-In (Ambion). Reverse transcription (50 °C for 30 min) was followed by a pre-amplification step (24 cycles: 95 °C for 15 s, 60 °C for 4 min) with a mixture of the probes below (each diluted to 0.2× standard working concentration) in a total volume of 9 μl. The product was diluted to a volume of 30 μl, and aliquots of 1 μl were then analysed using individual probes (final concentration = 1×) for a further 45 cycles using 7900 HT Fast Real-Time PCR system (Applied Biosystems).

The following probes from Applied Biosystems were used: P2rx1 Mm_00435460_m1; P2rx2 Mm_00462952_m1; P2rx3 Mm_00523699_m1; P2rx4 Mm_00501787_m1; P2rx5 Mm_00473677_m1; P2rx6 Mm_00440591_m1; P2rx7 Mm_01199500_m1; Glp1r Mm_00445292_m1; and Gapdh Mm_99999915_g1. Relative expression of each gene of interest was calculated by comparison to expression of the housekeeping gene Gapdh using a ΔCt method[54]. Undetermined CT values were assigned a value of 45 (the maximum number of cycles run) to enable statistical analysis.

**Drugs, chemicals and solutions**. Unless otherwise stated, all chemicals were purchased from Sigma-Aldrich. CNO, Ro51 and AR-C 118925XX were from Tocris (Biotechne), Exendin 9-39 was from Bachem. A Gq-DREADD expression plasmid was purchased from Addgene. Drugs for imaging and electrophysiology experiments were applied directly onto cells using a custom-made gravity-fed perfusion system. To reduce flow-induced artefacts, a constant flow of external solution was applied onto cells during baseline recordings and switched to a drug solution during drug application. Unless otherwise stated, recordings were performed at room temperature (20–24 °C).

Standard saline (138 buffer) contained (in mM): NaCl (138), KCl (4.5), HEPES (10.0), NaHCO₃ (4.2), NaH₂PO₄ (1.2), CaCl₂ (2.6), MgCl₂ (1.2); pH 7.4 with NaOH. The following concentrations of D-glucose were used: for ATP secretion experiments 0.1 mM, for electrophysiological experiments 1 mM, and for Fura-2 imaging experiments, 5 mM glucose. The Ringer's solution used in Ussing chamber experiments contained (in mM): NaCl (120), KCl (3.0), MgCl₂ (0.5), CaCl₂ (1.25), NaHCO₃ (23.0), and D-glucose (10.0), and constantly bubbled with carbogen (95% $O_2$/5% $CO_2$), pH 7.4 ± 0.2.

**Statistics**. Individual data points were represented on graphs with mean ± SEM or median ± interquartile range as indicated in the legends. Normally distributed data were analysed by Student's t test or ANOVA with Dunnett's multiple comparisons test, or one-sample t-test, as indicated in the figure legends. The threshold for statistical significance was $P < 0.05$. Statistical analysis was performed using GraphPad Prism software, and Adobe Illustrator was used to compile figures for presentation.

**Reporting summary**. Further information on experimental design is available in the Nature Research Reporting Summary linked to this article.

## Data availability

The source data underlying Figs. 1–6 and Supplementary Figure 1-5 are provided as a supplementary information Source Data file. All original raw data files are available from the corresponding author upon reasonable request. Unique biological materials (e.g. transgenic mice) are available for collaborations from the authors upon reasonable request. The code used for co-localisation and TIRF analysis is available at https://bitbucket.org/cwissmiff/travis/src under an Academic Free License (v3.0).

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

## Acknowledgements

This research was funded by a Wellcome Trust joint investigator award to FR/FMG (106262/Z/14/Z and 106263/Z/14/Z) and a joint MRC programme within the Metabolic Diseases Unit (MRC_MC_UU_12012/3). The MRL Histology and Biochemistry Assay Lab Core facilities received funding from the MRC Metabolic Diseases Unit [MRC_MC_UU_12012/5] and the Imaging Core through a Wellcome Trust Strategic Award [100574/Z/12/Z]. J.R. received a project support grant from the British Society for Neuroendocrinology. We thank Dan Drucker (Toronto) for the use of GLUTag cells and James Hockley for technical advice on single-cell qRT-PCR experiments.

## Author contributions

Immunohistochemistry, ATP secretion, qRT-PCR and nodose neuron co-cultures were performed by V.B.L. and E.A.O. V.B.L. performed sniffer patch experiments. J.R. performed Ussing chamber experiments. C.A.S. performed TIRF experiments and analysed the colocalization of VNUT with hormones. R.P and G.T. collected data at an early stage of the project. A.B.L. provided NeuroD1-Cre mice. Afferent nerve recordings were conducted by V.B.L., L.A.P. and D.C.B. F.M.G. and F.R. supervised the project and wrote the manuscript together with V.B.L. All authors analysed and/or interpreted data and approved the manuscript.

## Additional information

**Competing interests:** F.M.G. has consulted for Kallyope (New York) and the FR/FMG laboratories recieved industrial funding for other projects from MedImmune/AstraZeneca, LGC and Lilly, which has not influenced this study in any way. The remaining authors declare no competing interests.

