## [Peer Review File · Nature Communications]

Reviewers' comments:

Reviewer #1 (Remarks to the Author):

This paper is an interesting report of evidence for ATP as a mediator released from GLP-secreting cells. It uses a variety of impressive methods to acquire this evidence, but outcomes are indirect, and may not reflect what happens in the natural situation, so my question which is not answered in the paper is "Where is the evidence showing ATP release by a natural stimulus (found normally in the lumen) from native L-cells?"

Specific comments

- A more recent two-photon fluorescent probe for ATP and ADP has been developed that is markedly superior to quinacrine, which is subject to artifact and non-specific labeling, plus the drawbacks that the authors describe in Results. Confirmation of the observation is needed with more than one method, as this is a critical issue in the paper, especially the co-localization of ATP with other mediators as reported in Ref 22. No numerical compiled data are shown for quinacrine which is not acceptable.

- This is by no means the first report of ATP as a transmitter from EEC to sensory nerves. The following paper shows this and does so by more direct electrophysiological recording of the enteric neurons and muscle.

Gwynne RM, Bornstein JC. Local inhibitory reflexes excited by mucosal application of nutrient amino acids in guinea pig jejunum. *Am J Physiol Gastrointest Liver Physiol* 292: G1660–G1670, 2007

- I am concerned about the use of co-culture in this situation. Sensory neurons are well-known to change their properties (especially purinergic receptors) after culture*, and the trophic influence of co-culture with EEC or their surrogate is unknown. Therefore neither cell is likely to be in its natural configuration.

*Stebbing MJ, McLachlan EM, Sah P. Are there functional P2X receptors on cell bodies in intact dorsal root ganglia of rats? *Neuroscience*. 1998 Oct;86(4):1235-44.

- I don't understand why it isn't possible to study EEC-afferent neuron interactions in an intact preparation, more like the ISc experiments. This would provide much more direct evidence. Many groups have done this for ATP signaling, including at least five:

Wynn G, Rong W, Xiang Z, Burnstock G. Purinergic mechanisms contribute to mechanosensory transduction in the rat colorectum. *Gastroenterology*. 2003 Nov;125(5):1398-409.

Kirkup AJ, Booth CE, Chessell IP, Humphrey PP, Grundy D. Excitatory effect of P2X receptor activation on mesenteric afferent nerves in the anaesthetised rat. *J Physiol*. 1999 Oct 15;520 Pt 2:551-63.

Page AJ, O'Donnell TA, Blackshaw LA. P2X purinoceptor-induced sensitization of ferret vagal mechanoreceptors in oesophageal inflammation. *J Physiol*. 2000 Mar 1;523 Pt 2:403-11.

Zagorodnyuk VP1, Chen BN, Costa M, Brookes SJ. Mechanotransduction by intraganglionic laminar endings of vagal tension receptors in the guinea-pig oesophagus. *J Physiol*. 2003 Dec 1;553(Pt 2):575-87.

Shinoda M1, La JH, Bielefeldt K, Gebhart GF. Altered purinergic signaling in colorectal dorsal root ganglion neurons contributes to colorectal hypersensitivity. *J Neurophysiol*. 2010

Dec;104(6):3113-23.

- Are the increments in [ATP] in Fig 2 biologically significant? I would have expected a several fold increase such as the authors have shown elsewhere with GLP-1.

- The use of the Gq DREADD transgenic mouse and the stimulus of CNO is a concern, since the pathway of excitation secretion coupling may be quite different in pathway and potency in this situation compared to macronutrient-evoked EEC activation. It is very unclear to this reviewer why

a nutrient, like many the authors have studied, was not used. Likewise, the need to use AngII is obscure. The data from these experiments is complicated to interpret due to the recruitment of other pathways, in which case a more rigorous pharmacological approach is required.

Reviewer #2 (Remarks to the Author):

This study focuses on identifying that ATP is contained in L cell vesicles, that it is actively secreted from L cells, and that it may have functional consequences for paracrine signalling with epithelial cells and neuronal signalling with nearby nerve endings. This work provides a new paradigm in enteroendocrine signalling and combines leading-edge approaches to answer these questions from multiple angles. While I believe this represents an innovative and significant contribution to the field, I nonetheless have several questions for the authors. Overall though these questions may be considered relatively minor.

The authors show quinacrine staining in GLUTag cells and RFP-labelled mouse L cells. While this certainly looks punctate this data does not demonstrate that it is vesicular. Furthermore, ATP localisation could be assessed using fixed tissue through the use of antibodies against the Vesicular Nucleotide Transporter (VNUT) which transports ATP into vesicles. Such data that co-labelled for a vesicle marker (eg; GLP-1) would be unequivocal evidence that ATP is indeed contained in secretory vesicles rather than other similar structures like endosomes or lysosomes. In addition to this, the impact of the findings in this paper would be further strengthened if such staining could be provided in human gut tissue to demonstrate the potential clinical relevance of this system.

In the secretion experiments in Figure 2, over half the ATP is taken into cells within the 10 minutes that the secretion experiments in Fig 2B-D were conducted. This was only ~25% in the presence of POM-1. Were the secretion experiments conducted in the presence of POM-1 to reduce this loss of ATP from supernatants? It would appear to be an approach that would increase the fidelity of the secretion experiments.

Did the authors check that CNO did not cause ATP secretion in mock transfected cells?

In Fig 3C, is the scale bar correct? The response to CNO is ~.1nA but the size of responses in 3D are much larger.

Line 201-204. This suggestion does not seem to have evidence to support it. An alternative could just as easily be that ATP secreted from a nearby GLUTag cell has a paracrine effect on the ND. Why do the authors suggest something that sounds like a synaptic connection? Is a synaptic link physically observed between the images ND and GLUTag cells studied?

Regardless of this answer, does this data really provide evidence that L cell-derived ATP activates nodose ganglia? In reality this is just the co-culture of 2 cell types 1 of which secretes ATP and the other which is responsive to ATP. Whether this occurs in vivo is not answered with such an approach.

Why does suramin cause 75% of GLUTag cells to be activated? This is unexpected.

In 6d, there appears to be a delay in the ND cell responses compared to GLUTag cells activation. This delay in ND cells isn't seen in 6b. why is this? The size of the responses seems to be much bigger for ND cells in 6d vs 6b also, why?

Reviewer #3 (Remarks to the Author):

In this paper the authors investigate the role of ATP as a fast neurotransmitter in specialized sensory enteroendocrine L-cells (EECs) of the intestinal epithelium. In addition to an immortalized model colonic cell line, i.e. GLUTag cells, they also test primary EECs from adult mouse gut

epithelium. The authors use several assays including ATP measurements in cell supernatants, sniffer patches expressing purinergic P2X2R, and real-time imaging of quinacrine-loaded vesicles to show that ATP is released from stimulated EECs. They further show this stimulated ATP release can activate P2Y receptors on neighboring enterocytes so as to modulate the electrical properties of the isolated epithelium. Finally, they use co-cultures of GLUTag cells and dispersed nodose ganglion (ND) neurons, combined with ratiometric Ca²⁺ imaging, to demonstrate that stimulated ATP release from EECs can cause activation of nearby ND neurons. They suggest their data identifies a potential purinergic afferent signaling pathway for triggering vagal activation (and presumably centrally-mediated anorexic responses) via innervated EEC.

General Comments:

The authors use a nice combination of techniques, including specific cell labeling in transgenic mouse models, to show stimulated ATP release from EECs. However, this result is not entirely surprising given that ATP is often co-released with hormones and other neurotransmitters in a variety of endocrine and neuronal cells. A major goal of the study was to show ATP acts as a fast transmitter so as to activate the vagal pathway and thereby contribute to the anorexic action mediated by EECs. This is summarized in their Discussion lines 315-317 as follows: "This result suggests that ATP from L-cells acts as a fast transmitter, capable of activation local nerve endings expressing purinergic receptors such as sensory vagal afferent neurons of the nodose ganglion". Unfortunately, the purinergic receptors were inadequately characterized and therefore the data provided to support this thesis appear preliminary.

The latter conclusion is especially apparent when contrasted with other sensory afferent systems where ATP has been shown to act as a critical fast transmitter at peripheral receptors, e.g. carotid body (CB) O₂ receptors and gustatory taste receptors. In the CB, purinergic P2X_{2/3} receptors were immunolocalized to afferent nerve terminals innervating receptor cells, and fast ATP neurotransmission was demonstrated in CB cocultures (Prasad et al. 2001, *J Physiol* 537: 667-77; Nurse and Piskuric 2013, *Semin Cell Dev Biol* 24: 22-30). Most importantly, knockout of P2X_{2R} in transgenic mouse models, practically abolished the CB afferent sensory discharge to low O₂ as well as hyperventilatory response in such animals exposed to low O₂ (Rong et al. 2003, *J Neurosci.* 23: 11315-21). Similarly in the taste bud, genetic elimination of P2X₂ and P2X₃ receptors abolished taste responses in taste nerves and behavioral responses to various tastants (Finger et al. 2005, *Science* 310: 1495-9). Also, P2X₂ and P2X₃ receptors were immunolocalized to afferent terminals in the taste bud (Ishida et al. 2009, *J Comp Neurol* 514: 131-44), and selective blockade of postsynaptic P2X₃ receptors abolished taste transmission (Vandenbeuch et al 2015, *J Physiol.* 593: 1113-25). Together, these studies set the standard for a convincing demonstrating of ATP as a fast transmitter at peripheral sensory receptors.

Related to the present study, it should be noted that P2X receptors in sensory ganglia, including the ND ganglia, have been very well characterized by RT-PCR, immunohistochemical, pharmacological, and electrophysiological techniques (see Reviews from Burnstock and North labs). There is a wealth of information on P2X_{2/3R} (including desensitization kinetics, sensitivity to agents such as $\alpha\beta$ -methylene ATP and AF-353, etc), and therefore the authors should be able to apply these tools to characterize the P2X receptors on Glp1r-expressing ND neurons. They should also be able to immuno-localize the physiologically relevant subunits near EEC (L-cells) in situ. In fact, the ND cell bodies which were the focus of the present study are a long way from the EEC in vivo and therefore the precise P2X receptor localization at vagal terminals is central to the authors' hypothesis. P2X_{2/3} knockout models should also be employed to test whether or not the anorexic function of EEC is impaired in vivo when fast purinergic neurotransmission is abrogated, as demonstrated at taste bud and carotid body receptors.

Specific Comments:

1. Quinacrine accumulates in acidic compartments and partitions across a pH gradient. As such, it will accumulate in lysosomal granules which in turn can release ATP (see Sivaramakrishnan et al. 2012, *J Cell Science* 125: 4567-75; Liu et al. 2016, *Sci Reports* 6: 20903). Can the authors exclude lysosomes in EEC cells as a source of ATP release? At least this point should be discussed.

2. In several figures (3,4,5,6), concentrations of agonists/drugs are indicated in mM instead of μM as indicated in legends/text. In Fig. 1 scale bar reads mm instead of μm .
3. In Fig. 3e,f, ATP-like currents were detected in sniffer patches placed near fluorescent primary colonic L-cells following AngII stimulation. Were these currents blocked by suramin or PPADS?
4. In several experiments the authors could have quantified and compared response latencies between the stimulated cells and the ATP-sensitive target cells.
5. Why didn't the authors attempt co-cultures of ND neurons and primary cultures containing colonic L-cells? A positive outcome would have greatly strengthened their case. In this regard, a more detailed description of GLUTag cells-ND co-cultures would have been helpful. It is difficult to discern from the micrographs provided whether ND neural processes are present in these 48 hr co-cultures. This raises the question whether the Ca^{2+} signals recorded in ND neurons are based on soma-soma contacts with GLUTag cells. If not, and they are due to propagated action potentials originating in ND terminals near GLUTag cells, then are these Ca^{2+} signals sensitive to TTX?
6. It seems more efficient to merge Figs. 1 & 4.
7. In Fig. 2C, there is an outlier (around 20 nM ATP) among the CNO data points. Is significance still present with that outlier removed?
8. Figs. 6b,d show 2 GLUTag cells, one that responds to ATP but the other does not. How do the authors account for these differences. Also, in Figs. d,e, the integrated ND Ca^{2+} response to CNO-stimulation of GLUTag cells was inhibited by PPADS. Was the peak Ca^{2+} response similarly inhibited?

Author's response to Reviewers Comments:

Reviewer #1:

This paper is an interesting report of evidence for ATP as a mediator released from GLP-secreting cells. It uses a variety of impressive methods to acquire this evidence, but outcomes are indirect, and may not reflect what happens in the natural situation, so my question which is not answered in the paper is "Where is the evidence showing ATP release by a natural stimulus (found normally in the lumen) from native L-cells?"

Response: We have conducted additional experiments, using the sniffer-patch assay for ATP release, to address this issue. We have now tested nutrient receptor stimuli of colonic L-cells, including agonists for the free-fatty acid receptor 1 (FFA1) and the bile acid receptor (GPBAR-1), as well as nutrient stimuli, peptones. All three stimuli produce P2X2 currents in our sniffer-patch assay, though not in every sniffer patch tested, and these data are now included in a revised Figure 3f. The variability likely reflects differential receptor expression between individual colonic L-cells, which would be observed in single cell assays such as the sniffer-patch system.

Specific comments

- A more recent two-photon fluorescent probe for ATP and ADP has been developed that is markedly superior to quinacrine, which is subject to artifact and non-specific labeling, plus the drawbacks that the authors describe in Results. Confirmation of the observation is needed with more than one method, as this is a critical issue in the paper, especially the co-localization of ATP with other mediators as reported in Ref 22. No numerical compiled data are shown for quinacrine which is not acceptable.

Response: As suggested also by the other reviewers, we have added experiments investigating the co-localization of ATP-containing vesicles with vesicles containing GLP-1 (Figure 1h-j). As mentioned in the manuscript, we were unable to stain quinacrine and GLP-1 simultaneously. However, utilizing an indirect marker of ATP in vesicles, the vesicular nucleotide transporter (VNUT), we were able to demonstrate a partial colocalization of VNUT and GLP-1 in vesicular structures, thus supporting a mechanism of ATP co-release with peptide hormones from GLP-1 releasing cells.

The increase in quinacrine flashing events upon application of AngII had been quantified and the numbers had been given in the text. However, these were based on events observed in different cells statically incubated either in the presence or absence of AngII. For the resubmission of the manuscript we attempted to record an increase in flashing events from individual cells superfused successively without and with AngII, but failed to observe an increase in flashing events. Although we suspect that this reflects technical problems with our TIRF

setup, given that AngII increased ATP release when measured in the supernatant or by “sniffer-patches”, we have decided to only report quinacrine flashing seen under basal conditions in the current manuscript. We have, as requested by the reviewer, quantified quinacrine flashing under basal conditions and it is now reported in the figure legend (Figure 1f).

- *This is by no means the first report of ATP as a transmitter from EEC to sensory nerves. The following paper shows this and does so by more direct electrophysiological recording of the enteric neurons and muscle.*

Gwynne RM, Bornstein JC. Local inhibitory reflexes excited by mucosal application of nutrient amino acids in guinea pig jejunum. Am J Physiol Gastrointest Liver Physiol 292:G1660–G1670,2007

Response: As the reviewer suggested, this reference is now cited in the discussion. While Gwynne and Bornstein speculate that EECs are the likely source of mucosal ATP upon amino acid stimulation, there is, however, no evidence for this in their publication.

- *I am concerned about the use of co-culture in this situation. Sensory neurons are well-known to change their properties (especially purinergic receptors) after culture*, and the trophic influence of co-culture with EEC or their surrogate is unknown. Therefore neither cell is likely to be in its natural configuration.*

**Stebbing MJ, McLachlan EM, Sah P. Are there functional P2X receptors on cell bodies in intact dorsal root ganglia of rats? Neuroscience. 1998 Oct;86(4):1235-44.*

Response: We have carried out expression analysis of the various *P2rx* receptor subunits in nodose (ND) ganglion neurons in 3 different preparations: intact ganglia (most likely to represent ND neurons in their native state), acutely dissociated ND neurons, and dissociated neurons after 3 days in vitro (resembling the state of ND neurons utilized in co-culture imaging experiments). As shown in the new Supplemental Figure 5a, the profile of *P2rx* subunits was similar across all preparations, with *P2rx3* exhibiting the highest expression levels. However, expression of *P2rx2* and *3* were elevated in acutely dissociated samples compared with intact whole ND and after 3 days in vitro. In fact, the restoration of expression levels of all *P2rx* subunits after 3 days, to similar levels found in the intact ND ganglia, gives us confidence that the properties of ND neurons utilized in our co-culture imaging experiments will represent the conditions found in intact nodose ganglia.

- *I don't understand why it isn't possible to study EEC-afferent neuron interactions in an intact preparation, more like the ISc experiments. This would provide much more direct evidence. Many groups have done this for ATP signaling, including at least five: Wynn G, Rong W, Xiang Z, Burnstock G. Purinergic mechanisms contribute to mechanosensory transduction in the rat colorectum. Gastroenterology. 2003 Nov;125(5):1398-409.*

Kirkup AJ, Booth CE, Chessell IP, Humphrey PP, Grundy D. Excitatory effect of P2X receptor activation on mesenteric afferent nerves in the anaesthetised rat. J Physiol. 1999 Oct 15;520 Pt 2:551-63.

Page AJ, O'Donnell TA, Blackshaw LA. P2X purinoceptor-induced sensitization of ferret vagal mechanoreceptors in oesophageal inflammation. J Physiol. 2000 Mar 1;523 Pt 2:403-11.

Zagorodnyuk VP1, Chen BN, Costa M, Brookes SJ. Mechanotransduction by intraganglionic laminar endings of vagal tension receptors in the guinea-pig oesophagus. J Physiol. 2003 Dec 1;553(Pt 2):575-87.

Shinoda M1, La JH, Bielefeldt K, Gebhart GF. Altered purinergic signaling in colorectal dorsal root ganglion neurons contributes to colorectal hypersensitivity. J Neurophysiol. 2010 Dec;104(6):3113-23.

Response: We thank the reviewer for this suggestion and have established a collaboration with the group of Dr David Bulmer (added as new authors) to carry out electrophysiological recordings of afferent nerve activity from intact mouse colonic tissue preparations (Figure 6). We recorded electrical activity from nerves innervating the proximal colon, an area shown to be innervated by vagal afferents. These new results have been added to the manuscript.

- Are the increments in [ATP] in Fig 2 biologically significant? I would have expected a several fold increase such as the authors have shown elsewhere with GLP-1.

Response: To represent ATP secretion experiments in a more intuitive way, the units for ATP concentration were converted to ng/ml. Also, some ATP secretion experiments were repeated with the addition of an inhibitor of the enzyme that breaks down ATP, POM-1 (Figure 2b, d). With and without POM-1 present, we observed a doubling of ATP released following stimulation by known secretagogues of GLP-1 release, AngII and F/I/10G, and the specific agonist of Gq-DREADDs. The fold increase measured in vitro at a fixed time point is not, however, a good measure of the dynamic changes that would be observed around an actively secreting cell, as "basal" ATP levels in supernatants are strongly influenced by any ATP released from non-EECs and EECs that did not respond to the stimulus. Both basal and stimulated ATP levels are also influenced by rates of ATP disappearance through different pathways. Nevertheless, the 2-fold increase in supernatant ATP concentration is not so different from the ~3.5-fold increase in GLP-1 secretion (Figure 2f) observed under similar conditions. By contrast, our sniffer patch experiments indicate that ATP concentrations in the vicinity of an L-cell increase much more dramatically after L-cell stimulation than is apparent in the supernatant concentration measurements. In our new intact nerve recordings (Figure 6) we also found evidence that stimulation of ATP release from L-cells is sufficient to increase afferent nerve activity, which could be partially blocked by the broad-spectrum ATP receptor blocker PPADS.

The use of the Gq DREADD transgenic mouse and the stimulus of CNO is a concern,

since the pathway of excitation secretion coupling may be quite different in pathway and potency in this situation compared to macronutrient-evoked EEC activation. It is very unclear to this reviewer why a nutrient, like many the authors have studied, was not used. Likewise, the need to use AngII is obscure. The data from these experiments is complicated to interpret due to the recruitment of other pathways, in which case a more rigorous pharmacological approach is required.

Response: We deliberately chose AngII and the Gq-DREADD system, as methods that we were confident would specifically stimulate L-cells in mixed intestinal cultures without non-specifically recruiting other cell types. However, we have also now tested endogenous receptor coupling mechanisms known to trigger GLP-1 release from L-cells, including agonists for the free-fatty acid receptor 1 (FFA1 – Gq coupled), the bile acid receptor (GPBAR-1 – Gs coupled) and peptones (activating CASR plus likely other pathways) (Figure 3f). Targeting these receptors enhanced ATP release, confirming that the responses observed with AngII and CNO were not an artefact of the system.

Reviewer #2 (Remarks to the Author):

This study focuses on identifying that ATP is contained in L cell vesicles, that it is actively secreted from L cells, and that it may have functional consequences for paracrine signalling with epithelial cells and neuronal signalling with nearby nerve endings. This work provides a new paradigm in enteroendocrine signalling and combines leading-edge approaches to answer these questions from multiple angles. While I believe this represents an innovative and significant contribution to the field, I nonetheless have several questions for the authors. Overall though these questions may be considered relatively minor.

The authors show quinacrine staining in GLUTag cells and RFP-labelled mouse L cells. While this certainly looks punctate this data does not demonstrate that it is vesicular. Furthermore, ATP localisation could be assessed using fixed tissue through the use of antibodies against the Vesicular Nucleotide Transporter (VNUT) which transports ATP into vesicles. Such data that co-labelled for a vesicle marker (eg; GLP-1) would be unequivocal evidence that ATP is indeed contained in secretory vesicles rather than other similar structures like endosomes or lysosomes. In addition to this, the impact of the findings in this paper would be further strengthened if such staining could be provided in human gut tissue to demonstrate the potential clinical relevance of this system.

Response: We thank Reviewer 2 for this helpful suggestion and have carried out immunohistochemical experiments to localize VNUT and GLP-1 in GLUTag cells and primary human and mouse intestinal cultures (Figure 1 h-j). We found that VNUT and GLP-1 were at least in part colocalised in the same vesicular structures, supporting the idea that ATP is located in the same vesicular population as GLP-1 and other peptide hormones. To further demonstrate the potential relevance of our

study in humans, we have also measured ATP secretion from primary colonic cultures prepared from human biopsy samples (Figure 2g).

In the secretion experiments in Figure 2, over half the ATP is taken into cells within the 10 minutes that the secretion experiments in Fig 2B-D were conducted. This was only ~25% in the presence of POM-1. Were the secretion experiments conducted in the presence of POM-1 to reduce this loss of ATP from supernatants? It would appear to be an approach that would increase the fidelity of the secretion experiments.

Response: We have repeated all ATP secretion experiments in which POM-1 was not originally included, and have added these data to a revised Figure 2. We have also left the non-POM1 treated GLUTag secretion data in this Figure as this demonstrates that POM-1 did not alter the overall response profile to the agonists tested.

Did the authors check that CNO did not cause ATP secretion in mock transfected cells?

Response: Yes, we did check that CNO does not stimulate ATP secretion in untransfected (Lipofectamine applied but no cDNA was included) cells. The control groups are shown in Figure 2c and d.

In Fig 3C, is the scale bar correct? The response to CNO is ~.1nA but the size of responses in 3D are much larger.

Response: We thank the Reviewer for noticing this error, introduced because we resized the scale bar but did not change the label. We have now corrected the figure.

Line 201-204. This suggestion does not seem to have evidence to support it. An alternative could just as easily be that ATP secreted from a nearby GLUTag cell has a paracrine effect on the ND. Why do the authors suggest something that sounds like a synaptic connection? Is a synaptic link physically observed between the images ND and GLUTag cells studied?

Response: The reviewer is correct that our data do not themselves indicate whether the transmission was synaptic or paracrine, although the synaptic idea has been suggested by the work of other groups (Bohorquez et al., 2015. J Clin Invest 125(2):782.6; Bellono et al., 2017. Cell 170:185-198). We have altered the discussion of our data accordingly.

Regardless of this answer, does this data really provide evidence that L cell-derived ATP activates nodose ganglia? In reality this is just the co-culture of 2 cell types 1 of

which secretes ATP and the other which is responsive to ATP. Whether this occurs in vivo is not answered with such an approach.

Response: The reviewer is correct that our original experiments did not quantify the importance of purinergic transmission between L-cells and the afferent vagus in vivo. As suggested by another reviewer, however, we have now recorded from afferent nerves during stimulation of L-cells in the intact mouse colon, which revealed AngII-triggered increases in nerve activity that were blocked by the purinergic inhibitor PPADS. These new data are included in the new Figure 6, and provide support for a role for purinergic signaling between L-cells and afferent nerves in vivo. Of course this might still be a paracrine rather than a synaptic transmission, however, the point of the manuscript is to establish additional signalling moieties between enteroendocrine cells and the brain.

Why does suramin cause 75% of GLUTag cells to be activated? This is unexpected.

Response: We do not have an answer to this question, and suspect it may an off-target effect.

In 6d, there appears to be a delay in the ND cell responses compared to GLUTag cells activation. This delay in ND cells isn't seen in 6b. why is this? The size of the responses seems to be much bigger for ND cells in 6d vs 6b also, why?

Response: Overall, response magnitudes were variable between cells, and no consistent differences were observed between experiments. We were also curious about the time delay observed in some experiments, and had already mentioned this in the discussion. We have also now included discussion of the possibility that we cannot exclude a paracrine interaction in some or all cases. The relevant discussion now reads:

“However, local paracrine signalling cannot be excluded, as in some cases we noticed neuronal responses that were “delayed” relative to those seen in GLUTag cells monitored in the same optical field”.

Reviewer #3 (Remarks to the Author):

In this paper the authors investigate the role of ATP as a fast neurotransmitter in specialized sensory enteroendocrine L-cells (EECs) of the intestinal epithelium. In addition to an immortalized model colonic cell line, i.e. GLUTag cells, they also test primary EECs from adult mouse gut epithelium. The authors use several assays including ATP measurements in cell supernatants, sniffer patches expressing purinergic P2X2R, and real-time imaging of quinacrine-loaded vesicles to show that ATP is released from stimulated EECs. They further show this stimulated ATP release can activate P2Y receptors on neighboring enterocytes so as to modulate the electrical properties of the isolated epithelium. Finally, they use co-cultures of GLUTag cells and

dispersed nodose ganglion (ND) neurons, combined with ratiometric Ca²⁺ imaging, to demonstrate that stimulated ATP release from EECs can cause activation of nearby ND neurons. They suggest their data identifies a potential purinergic afferent signaling pathway for triggering vagal activation (and presumably centrally-mediated anorexic responses) via innervated EEC.

General Comments:

The authors use a nice combination of techniques, including specific cell labeling in transgenic mouse models, to show stimulated ATP release from EECs. However, this result is not entirely surprising given that ATP is often co-released with hormones and other neurotransmitters in a variety of endocrine and neuronal cells. A major goal of the study was to show ATP acts as a fast transmitter so as to activate the vagal pathway and thereby contribute to the anorexic action mediated by EECs. This is summarized in their Discussion lines 315-317 as follows: "This result suggests that ATP from L-cells acts as a fast transmitter, capable of activation local nerve endings expressing purinergic receptors such as sensory vagal afferent neurons of the nodose ganglion". Unfortunately, the purinergic receptors were inadequately characterized and therefore the data provided to support this thesis appear preliminary.

The latter conclusion is especially apparent when contrasted with other sensory afferent systems where ATP has been shown to act as a critical fast transmitter at peripheral receptors, e.g. carotid body (CB) O₂ receptors and gustatory taste receptors. In the CB, purinergic P2X_{2/3} receptors were immunolocalized to afferent nerve terminals innervating receptor cells, and fast ATP neurotransmission was demonstrated in CB cocultures (Prasad et al. 2001, J Physiol 537: 667-77; Nurse and Piskuric 2013, Semin Cell Dev Biol 24: 22-30). Most importantly, knockout of P2X_{2R} in transgenic mouse models, practically abolished the CB afferent sensory discharge to low O₂ as well as hyperventilatory response in such animals exposed to low O₂ (Rong et al. 2003, J Neurosci. 23: 11315-21). Similarly in the taste bud, genetic elimination of P2X₂ and P2X₃ receptors abolished taste responses in taste nerves and behavioral responses to various tastants (Finger et al. 2005, Science 310: 1495-9). Also, P2X₂ and P2X₃ receptors were immunolocalized to afferent terminals in the taste bud (Ishida et al. 2009, J Comp Neurol 514: 131-44), and selective blockade of postsynaptic P2X₃ receptors abolished taste transmission (Vandenbeuch et al 2015, J Physiol. 593: 1113-25). Together, these studies set the standard for a convincing demonstrating of ATP as a fast transmitter at peripheral sensory receptors. Related to the present study, it should be noted that P2X receptors in sensory ganglia, including the ND ganglia, have been very well characterized by RT-PCR, immunohistochemical, pharmacological, and electrophysiological techniques (see Reviews from Burnstock and North labs). There is a wealth of information on P2X_{2/3R} (including desensitization kinetics, sensitivity to agents such as $\alpha\beta$ -methylene ATP and AF-353, etc), and therefore the authors should be able to apply these tools to characterize the P2X receptors on Glp1r-expressing ND neurons. They should also be able to immuno-localize the physiologically relevant subunits near EEC (L-cells) in situ. In fact, the ND cell bodies which were the focus of the present study are a long way from the EEC in vivo and therefore the precise P2X receptor localization at vagal terminals is central to the authors' hypothesis. P2X_{2/3} knockout models should also be employed to test whether or not the anorexic function

of EEC is impaired in vivo when fast purinergic neurotransmission is abrogated, as demonstrated at taste bud and carotid body receptors.

Response: We thank Reviewer 3 for this suggestion and have carried out experiments examining which ATP receptor subunits are expressed in ND neurons (Supplemental Figure 5). We have also tested the selective P2X3 and P2X2/3 selective agonist AF-353, which according to our expression analysis was the most abundant ATP receptor subunit expressed in ND neurons, and found a range of effectiveness in blocking ATP responses (100% to no blockade). This variability likely reflects the heterogeneity of ND neurons. We have also performed additional experiments on intact colon/nerve preparations showing that purinergic transmission occurs in the whole gut. Whilst we agree that there are many additional tools that could be used to further tease out the molecular pathway, we suggest that this level of detail is beyond the scope of the current manuscript.

Specific Comments:

1. Quinacrine accumulates in acidic compartments and partitions across a pH gradient. As such, it will accumulate in lysosomal granules which in turn can release ATP (see Sivaramakrishnan et al. 2012, J Cell Science 125: 4567-75; Liu et al. 2016, Sci Reports 6; 20903). Can the authors exclude lysosomes in EEC cells as a source of ATP release? At least this point should be discussed.

Response: We cannot exclude that some of the observed quinacrine staining might be in lysosomes, and have not co-stained for lysosomal markers, but our new data showing that VNUT staining overlaps with GLP-1 staining strongly supports the concept that ATP is concentrated in GLP-1 containing vesicles. In view of the finding that ATP secretion was stimulated in response to a range of stimuli known to increase GLP-1 release, we believe it is most likely that the ATP released by these stimuli arises from the peptidergic vesicles. It seems unlikely that ATP contained in a lysosomal pool would respond to the same range of stimuli that trigger GLP-1 secretion.

2. In several figures (3,4,5,6), concentrations of agonists/drugs are indicated in mM instead of μ M as indicated in legends/text. In Fig. 1 scale bar reads mm instead of μ m.

Response: We have attempted to correct this problem, which appeared during the uploading and pdf-conversion of figures during the submission process. We hope this problem is corrected for the resubmitted version of the manuscript.

3. In Fig. 3e,f, ATP-like currents were detected in sniffer patches placed near fluorescent primary colonic L-cells following AngII stimulation. Were these currents blocked by suramin or PPADS?

Response: We have run control experiments on sniffer patches to test whether the currents elicited by exogenous ATP application are blocked by suramin (Supplemental Figure 2c-d). We have also tested that currents produced in response to AngII are also blocked by PPADS (n=2), as shown below (same cell, 2nd trace 20 min after PPADS treatment).

4. In several experiments the authors could have quantified and compared response latencies between the stimulated cells and the ATP-sensitive target cells.

Response: Although we could extract latencies we do not believe that this would give an insight into the connectivity of the underlying signaling events. The short circuit currents recorded in the Ussing chamber experiments are likely dominated by enterocyte activity and are delayed relative to the initial ATP release. In the co-culture experiments we have no evidence that the L-cells visible in the optical field are the major contributors to the ND activity in the optical field (other L-cells outside the optical field might contribute). The new experiments with the ex vivo colonic preparation indicate a slow delayed purinergic response as well.

5. Why didn't the authors attempt co-cultures of ND neurons and primary cultures containing colonic L-cells? A positive outcome would have greatly strengthened their case. In this regard, a more detailed description of GLUTag cells-ND co-cultures would have been helpful. It is difficult to discern from the micrographs provided whether ND neural processes are present in these 48 hr co-cultures. This raises the question whether the Ca²⁺ signals recorded in ND neurons are based on soma-soma contacts with GLUTag cells. If not, and they are due to propagated action potentials originating in ND terminals near GLUTag cells, then are these Ca²⁺ signals sensitive to TTX?

Response: We thank the reviewer for this suggestion and have conducted additional experiments in cocultures of primary intestinal epithelial cells with ND neurons (Figure 5g-i). We have not attempted to visualise the connections between the L-cells and ND neurons, and agree with the reviewer that the types of contacts between cultured GLP-1 releasing cells and ND neurons may not reflect true synapses as the factors responsible for synapse formation may be missing. TTX is also not necessarily a useful tool here, because L-cells also fire action potentials

dependent on voltage gated Na^+ and Ca^{2+} channels. We can, however, confirm that similar to what has been reported previously (Bohorquez et al., 2015. J Clin Invest 125(2):782.6), ND neurons and L-cells do form outgrowth projections towards one another, possibly neuronal/neuropod connections, after 48h co-culture.

6. It seems more efficient to merge Figs. 1 & 4.

Response: We have now added additional figures and merged others.

7. In Fig. 2C, there is an outlier (around 20 nM ATP) among the CNO data points. Is significance still present with that outlier removed?

Response: Yes, without this point in question, ATP levels in the CNO-treated Gq-DREADD transfected GLUTags are still significantly different from the CNO-treated untransfected GLUTags and basal-treated transfected cells, as shown below (one-way ANOVA, $*p < 0.05$, $**p < 0.01$). We have run a non-regression analysis to determine if this point in question should be counted as an outlier but this test failed to remove this point. Rather than remove a data point for no scientific reason, we have chosen to include this point in our final results.

8. Figs. 6b,d show 2 GLUTag cells, one that responds to ATP but the other does not. How do the authors account for these differences. Also, in Figs. d,e, the integrated ND

Ca²⁺ response to CNO-stimulation of GLUTag cells was inhibited by PPADS. Was the peak Ca²⁺ response similarly inhibited?

Response: We have already reported that only 89/123 GLUTag cells responded to ATP with an elevation in intracellular Ca²⁺ (page 8). We suspect there may be differential expression of ATP receptors on GLUTag cells, and are not surprised that not all cells behaved identically. Heterogeneity between individual L-cells was also a major feature of primary L-cells examined by single cell RNA sequencing from the small intestine (Glass LL et al., 2017. Mol Metab, 6(10):1296-1303) and we are in the process of collecting similar data for colonic L-cells.

We observed a significant inhibition by PPADS of the peak Ca²⁺ response amplitude as well as the integrated response in ND neurons co-cultured with Gq-transfected GLUTag cells, as shown below (paired t-test, **p<0.01), but opted to present the integrated area in our figure because the shape as well as the peak of the Ca²⁺ responses were affected by PPADS inhibition.

Reviewers' comments:

Reviewer #1:

Remarks to the Author:

The authors have done a fine job of improving the manuscript in the light of the reviewers' comments. The impact of the paper is mainly considerably improved. There is a problem, however, with the new experiments that have been performed. These have in fact detracted from the other improvements to the paper. They were poorly thought through and were performed in an unvalidated setting with inadequate controls. The experiments in question are those done by Dr Bulmer, who is a new collaborator on the paper. These are recordings from afferent fibres as they exit the colon en route to the spinal cord or vagus nerve. The investigators used AngII to activate L-cells, and thus release mediators from them. The big problem here is that the afferent nerves themselves possess AT receptors, so the effects of AngII observed are almost certainly direct upon their endings, and not upstream on L-cells that in turn release ATP onto them. This is a pretty obvious mistake to most readers, so the paper will lose some of its overall credibility. I suggest the data in question are removed and the shortcoming of the paper acknowledged. Although this still leaves the paper weaker than it could be, it is better than including misleading data.

Reviewer #2:

Remarks to the Author:

The authors have gone to extensive lengths to address all of the reviewer concerns including my own. Many of these concerns were overlapping between all 3 reviewers. I feel that this paper is now much improved from the original version and any doubts I had on specific aspects presented in the original paper no longer exist.

While the point was made by other reviewers that purinergic signalling has been shown previously in many cell types, I must make the point here that this has never been shown for any enteroendocrine cell type. The addition of the recordings in Figure 6 provide an additional layer of functional context to this original finding of L cell purinergic signalling.

I feel this is an important piece of work that is deserving of being presented to the field in a journal of this calibre.

Reviewer #3:

Remarks to the Author:

This revised manuscript contains some significant additions that result in a greatly improved version and for which I complement the authors. For example, the new Figure 6 showing ex vivo afferent signaling in intact colon together with PPADs inhibition of AngII response is particularly noteworthy. Furthermore, Fig. 5g showing PPADs block or partial block of Ca²⁺ signaling in ND neurons in primary cocultures of L cells expressing Gq-DREADD, and measurements of ATP secretion from L cells following stimulation of endogenous receptors represent substantive additions. Although, VNUT immunostaining of GLP-1 secreting cells in mouse ileum and human colonic cultures was also a welcomed addition, this reviewer would have liked to see P2X₂/3R immunostaining of ND nerve endings near immunolabelled L-cells in the intact colon. While Supp. Fig 5 does show high P2rx and P3rx mRNA expression in whole NG ganglia and cultures of ND cells, this information is not new and these samples contain heterogenous neuronal populations. PCR analysis of these subunits in identified GLP1R or NPY2R neurons from the nodose ganglia would have been far more convincing, especially in view of the broad variability in responses of P2X receptor antagonists to exogenous ATP, as shown in Supp. Fig. 5.

Minor comment:

In Methods section of ex vivo recordings, the authors should indicate the pH of extracellular solution.

Author's response to Reviewers Comments:

We appreciate the reviewers' comments to our revised manuscript and submit a revised version with the latest reviewers' comments and suggestions. Below are our responses to the specific reviewers comments.

Reviewers' comments:

Reviewer #1 (Remarks to the Author):

The authors have done a fine job of improving the manuscript in the light of the reviewers' comments. The impact of the paper is mainly considerably improved. There is a problem, however, with the new experiments that have been performed. These have in fact detracted from the other improvements to the paper. They were poorly thought through and were performed in an unvalidated setting with inadequate controls. The experiments in question are those done by Dr Bulmer, who is a new collaborator on the paper. These are recordings from afferent fibres as they exit the colon en route to the spinal cord or vagus nerve. The investigators used AngII to activate L-cells, and thus release mediators from them. The big problem here is that the afferent nerves themselves possess AT receptors, so the effects of AngII observed are almost certainly direct upon their endings, and not upstream on L-cells that in turn release ATP onto them. This is a pretty obvious mistake to most readers, so the paper will lose some of its overall credibility. I suggest the data in question are removed and the shortcoming of the paper acknowledged. Although this still leaves the paper weaker than it could be, it is better than including misleading data.

Response: We fully agree that the expression of AngII receptors on the afferent fibers themselves presents a challenge for these experiments. As discussed with the editors we have moved the afferent fiber recordings to the Supplementary Data section and have revised our discussion of this section as suggested. As also discussed with the editors and in agreement with comments from the other reviewers we have, however, still included these experiments in the manuscript, as the later component of the response is PPADS sensitive, whilst the immediate response likely arising from AT1A activation on the neurons, is not; this is consistent with a P2Y dependent secondary phase of activity, but we agree that this is relatively weak, as we cannot conclude unambiguously that this depends on ATP-release from L-cells and we have discussed this accordingly.

Reviewer #2 (Remarks to the Author):

The authors have gone to extensive lengths to address all of the reviewer concerns including my own. Many of these concerns were overlapping between all 3 reviewers. I feel that this paper is now much improved from the original version and any doubts I had on specific aspects presented in the original paper no longer exist.

While the point was made by other reviewers that purinergic signalling has been shown previously in many cell types, I must make the point here that this has never been shown

for any enteroendocrine cell type. The addition of the recordings in Figure 6 provide an additional layer of functional context to this original finding of L cell purinergic signalling.

I feel this is an important piece of work that is deserving of being presented to the field in a journal of this calibre.

Response: We thank Reviewer 2 for their assessment.

Reviewer #3 (Remarks to the Author):

This revised manuscript contains some significant additions that result in a greatly improved version and for which I complement the authors. For example, the new Figure 6 showing ex vivo afferent signaling in intact colon together with PPADs inhibition of AngII response is particularly noteworthy. Furthermore, Fig. 5g showing PPADs block or partial block of Ca²⁺ signaling in ND neurons in primary cocultures of L cells expressing Gq-DREADD, and measurements of ATP secretion from L cells following stimulation of endogenous receptors represent substantive additions. Although, VNUT immunostaining of GLP-1 secreting cells in mouse ileum and human colonic cultures was also a welcomed addition, this reviewer would have liked to see P2X_{2/3}R immunostaining of ND nerve endings near immunolabelled L-cells in the intact colon.

Response: We have attempted immunostaining for P2X₃, the most abundant P2X receptor subtype present in nodose neurons, in mouse colonic sections as suggested by Reviewer 3. We tested two commercially available antibodies (Alomone APR-016; Neuromics GP10108) and combined this with Neurofilament 200 (NF200; Abcam ab4680) immunostaining to identify nerve endings. Whilst some P2X₃ staining appeared to overlap with NF200 and some punctate staining appeared to be detectable near L-cells, there was extraneous P2X₃ staining and/or background staining in sections that made it difficult to clearly localize P2X₃-positive fibers in an intact colonic preparation. In parallel, we also stained for P2X₃ in dissociated nodose neurons and confirmed the punctate pattern of P2X₃ staining (Figure 6c and d) especially in processes extending from the cell body. We have also added functional data demonstrating P2X₃-mediated ATP signaling between enteroendocrine cells and nodose neurons by utilizing the P2X₃ and P2X_{2/3}-specific blocker Ro51 (Figure 6f-h). As the mRNA expression for P2X₃ was relatively similar in acutely isolated and cultured nodose neurons (Figure 6a) we think that the immunohistochemical and functional in vitro data clearly demonstrate a role of P2X₃ in GLP1R expressing (and possibly other) nodose neurons for signals received from L-cells

While Supp. Fig 5 does show high P2rx and P3rx mRNA expression in whole NG ganglia and cultures of ND cells, this information is not new and these samples contain heterogenous neuronal populations. PCR analysis of these subunits in identified GLP1R or NPY2R neurons from the nodose ganglia would have been far more convincing, especially in view of the broad variability in responses of P2X receptor antagonists to exogenous ATP, as shown in Supp. Fig. 5.

Response: We have added qRT-PCR experiments performed on single nodose neurons as suggested by the reviewer (Figure 6b). We have identified GLP1R-expressing neurons by utilizing a *Glp1r*-Cre mouse line (Richards et al., 2014, Diabetes) crossed with a reporter mouse and confirmed GLP1R-expression by screening for *Glp1r*. There was heterogeneity in the expression profile of P2rx subunits, but *P2rx3* was present in all nodose neurons selected and its expression was the highest of all subunits screened for (*P2rx1-7*).

Minor comment:

In Methods section of ex vivo recordings, the authors should indicate the pH of extracellular solution.

Response: This information has now been added in the Methods section.

REVIEWERS' COMMENTS:

Reviewer #3 (Remarks to the Author):

The manuscript is substantially improved with the addition of new experiments. The authors have satisfactorily addressed all my main concerns.

REVIEWERS' COMMENTS:

Reviewer #3 (Remarks to the Author):

The manuscript is substantially improved with the addition of new experiments. The authors have satisfactorily addressed all my main concerns.

Response: Many thanks.